# Targeting the Achilles’ Heel of Multidrug-Resistant *Staphylococcus aureus* by the Endocannabinoid Anandamide

**DOI:** 10.3390/ijms23147798

**Published:** 2022-07-14

**Authors:** Ronit Vogt Sionov, Shreya Banerjee, Sergei Bogomolov, Reem Smoum, Raphael Mechoulam, Doron Steinberg

**Affiliations:** 1Biofilm Research Laboratory, Institute of Biomedical and Oral Sciences, Faculty of Dentistry, Hadassah Medical School, The Hebrew University of Jerusalem, Jerusalem 9112102, Israel; shreya.banerjee@mail.huji.ac.il (S.B.); sergei.bogomolov@mail.huji.ac.il (S.B.); dorons@ekmd.huji.ac.il (D.S.); 2Institute for Drug Research, Faculty of Medicine, The Hebrew University of Jerusalem, Jerusalem 9112102, Israel; reems@ekmd.huji.ac.il (R.S.); raphaelm@ekmd.huji.ac.il (R.M.)

**Keywords:** anandamide, antibiotic resistance, autolysins, endocannabinoids, drug sensitization, *Staphylococcus aureus*, transmembrane transport, wall teichoic acid

## Abstract

Antibiotic-resistant *Staphylococcus aureus* is a major health issue that requires new therapeutic approaches. Accumulating data suggest that it is possible to sensitize these bacteria to antibiotics by combining them with inhibitors targeting efflux pumps, the low-affinity penicillin-binding protein PBP2a, cell wall teichoic acid, or the cell division protein FtsZ. We have previously shown that the endocannabinoid Anandamide (N-arachidonoylethanolamine; AEA) could sensitize drug-resistant *S. aureus* to a variety of antibiotics, among others, through growth arrest and inhibition of drug efflux. Here, we looked at biochemical alterations caused by AEA. We observed that AEA increased the intracellular drug concentration of a fluorescent penicillin and augmented its binding to membrane proteins with concomitant altered membrane distribution of these proteins. AEA also prevented the secretion of exopolysaccharides (EPS) and reduced the cell wall teichoic acid content, both processes known to require transporter proteins. Notably, AEA was found to inhibit membrane ATPase activity that is necessary for transmembrane transport. AEA did not affect the membrane GTPase activity, and the GTPase cell division protein FtsZ formed the Z-ring of the divisome normally in the presence of AEA. Rather, AEA caused a reduction in murein hydrolase activities involved in daughter cell separation. Altogether, this study shows that AEA affects several biochemical processes that culminate in the sensitization of the drug-resistant bacteria to antibiotics.

## 1. Introduction

*Staphylococcus aureus* is a nosocomial Gram-positive bacterium that is a common pathogen causing various kinds of infectious diseases [1,2,3]. It can cause mild infections on skin and soft tissues, but it can also develop into more serious and even life-threatening infections such as sepsis, pneumonia, osteomyelitis, and endocarditis [4]. *S. aureus* infections frequently involve biofilm formation on biotic materials such as skin, soft tissues, connective tissue, cardiac valves, and mucus [1,5,6], but can also form biofilms on abiotic medical devices such as prosthesis, catheters, implants, and stents [7,8,9,10]. Infections caused by *S. aureus* are often difficult to treat due to the bacteria’s rapid acquisition of various antibiotic-resistant mechanisms [3,11,12,13,14].

Many mechanisms can lead to antibiotic resistance in bacteria [11,12,13,14,15,16]. The most common mechanisms involve (i) expression of enzymes that inactivates antibiotics by either cleavage or modification of the antibiotics; (ii) alterations of the target molecule resulting in reduced affinity for the antibiotics; (iii) elevated expression of efflux pumps that extrude the antibiotics, thereby reducing their intracellular concentration; (iv) decreased membrane permeability, thus preventing the drug from reaching the target site; (v) altered metabolic state of the bacteria; (vi) appearance of persister cells that have adapted transient resistance mechanisms; (vii) induction of antibiotic-tolerant cells; and (viii) biofilm formation.

Explicitly, common resistance mechanisms in *S. aureus* include (i) overexpression of β-lactamases that cleave β-lactam antibiotics, leading to their inactivation [17]; (ii) overexpression of low-affinity penicillin-binding proteins such as PBP2a that are encoded by *mecA* present in the staphylococcal chromosomal cassette *SSCmec* element, thus conferring β-lactam resistance [13]; and (iii) overexpression of a variety of efflux pumps, resulting in resistance to a large range of antibiotics (e.g., NorA-C, AbcA, MdeA, QacA/B, and Opp1C) [3,13,14,18,19].

Embedded in a biofilm surrounded by an extracellular matrix, the bacteria are more protected from environmental stress stimuli, evade immune defense mechanisms, and exhibit increased resistance to antibiotics [20,21]. The extracellular matrix hampers the penetration of antibiotics, thereby preventing them from reaching the biofilm-embedded bacteria [12,22,23]. The lower metabolic activity of the sessile bacteria in biofilms makes them less sensitive to antibiotics whose action depends on cell division (e.g., β-lactam antibiotics) or active metabolism (e.g., tetracycline) [24,25]. Biofilm-embedded bacteria often express elevated levels of efflux pumps in comparison with planktonic growing bacteria, thereby strengthening the resistant phenotype [26,27]. Thus, antimicrobial regimes should include compounds that target biofilms. 

Many efforts have been invested to find the Achilles’ heel that can be targeted to overcome antibiotic resistance, especially in methicillin-resistant *S. aureus* (MRSA) and multidrug-resistant *S. aureus* (MDRSA) [6,11,28,29,30]. β-lactamase inhibitors such as clavulanic acid, sulbactam, and avibactam, increase the half-life of β-lactam antibiotics, thereby increasing their efficacy [31]. Methicillin was developed to resist the degradation by β-lactamase, but drug resistance appeared due to overexpression of PBP2a and PBP4 [32,33]. The cholesterol-lowering statin drug Zaragozic acid was found to sensitize MRSA to β-lactam antibiotics by disrupting functional membrane microdomains with resulting inactivation of PBP2a [34]. PBP2a can also be inactivated by the β-lactam antibiotic ceftaroline [35]. 

Another approach to sensitize MRSA to β-lactams is the combined treatment with inhibitors of the cell-division protein FtsZ that form the Z-ring divisome during cell division [36]. The penicillin-binding proteins PBP1, PBP2, PBP3, and PBP4 colocalize with FtsZ at the septum where they contribute to new cell wall formation [36]. The FtsZ inhibitor TXA707 disrupts septum formation, resulting in the relocalization of FtsZ away from midcell and membrane redistribution of the PBP proteins that are now made available for inhibition by the β-lactam oxacillin [36]. This study also showed that growth inhibition allows MRSA to respond to β-lactam antibiotics.

Furthermore, targeting one of the steps of cell wall teichoic acid (WTA) biosynthesis could sensitize MRSA to β-lactams even in the presence of PBP2a [37,38,39,40]. WTA affects several bacterial processes including regulation of cell division, protection from osmotic stress, and promotion of biofilm formation and host colonization [41,42,43]. Additionally, the cell wall component lipoteichoic acid (LTA) affects susceptibility to β-lactams [44,45,46] and regulates cell division and autolytic activity [45,46,47].

We have previously documented that the endocannabinoid anandamide (AEA; N-arachidonoylethanolamine), a lipid active compound acting as a retrograde neurotransmitter in the brain [48], could sensitize MRSA and MDRSA to various antibiotics including ampicillin, methicillin, gentamicin, tetracycline, and norfloxacin [49,50]. In addition, AEA inhibits biofilm formation of various *S. aureus* species including MRSA and MDRSA [49,50,51]. Combined with methicillin, it could demolish preformed biofilms [49]. As a single agent, AEA does not kill the bacteria, but rather induces an immediate growth arrest that is overcome with time [49]. AEA also induces an immediate membrane depolarization [49]. Importantly, it was observed that AEA leads to intracellular accumulation of various efflux pump substrates such as ethidium bromide, DAPI, and the antibiotic norfloxacin [49]. These data suggest that AEA acts as an efflux pump inhibitor, or alternatively, it affects transmembrane solute transport by different means. It was therefore interesting to study which biochemical processes are affected by AEA. The present study specifically looked at processes that are known to affect biofilm development and antibiotic sensitivity. These include penicillin-binding proteins and their membrane distribution, membrane fluidity and composition, exopolysaccharide (EPS) and extracellular DNA (eDNA) production, cell wall teichoic acid content, membrane ATPase activity, Z-ring divisome formation, and murein hydrolase activities. 

## 2. Results

### 2.1. The Combined Treatment of Anandamide and Methicillin Reduced the Viability of MDRSA 

Previous studies showed that anandamide (AEA) could sensitize both MRSA and MDRSA to various antibiotics including the penicillinase-resistant penicillin analog methicillin (MET) [49,51]. A major question was whether the combined treatment of AEA with MET was bacteriostatic or bactericidal. To this end, the growth of the MDRSA CI-M bacteria in the absence or presence of the agents was followed by counting colony-forming units (CFUs), which distinguish between the total numbers of live/dead bacteria measured by optical density (OD) and viable bacteria that can form colonies. In parallel, we performed a time-course study examining the relative ATP levels at various time points after treatment initiation using the BacTiter^Glo^ viability assay, in addition to measuring changes in OD. In accordance with previous studies [49], when analyzing planktonic growth by following the OD for a period of 8 h, AEA treatment led to retarded planktonic growth in comparison with control and methicillin-treated bacteria, and the combined treatment of AEA with MET prevented the bacterial growth during the tested period (Figure 1A). When looking specifically at the combined AEA/MET treatment (Figure 1B), an initial subtle increase in the OD was discerned, which returned to a level lower than the initial one after an 8 h incubation (Figure 1B), suggesting reduced viability over time. The OD of the combined AEA/MET treatment after a 24 h incubation was similar to that after 8 h (data not shown), suggesting a prolonged bacteriostatic effect following the initial bactericidal effect. 

When counting the CFUs after the indicated incubation periods, a strong reduction in the ability to form colonies was seen with AEA when compared with control bacteria, although there was a slight increase in CFUs over time when compared with time zero (Figure 1C). This goes along with previous studies showing a growth inhibitory activity of AEA that is overcome with time [49]. MET treatment reduced the CFUs in comparison with control bacteria (Figure 1C), which was more profound than the recorded changes in OD (Figure 1A). The number of CFUs, however, increased with time in the presence of MET (Figure 1C), which is explained by the MET-resistant phenotype of MDRSA CI-M [49]. However, when AEA was combined with MET, there was a gradual decrease in the CFUs reaching a 85–90% reduction after a 8 h incubation (Figure 1D), suggesting that the combined treatment reduces bacterial viability. After 24 h, the CFU of the combined treatment was slightly higher than that after 8 h, but still lower than the initial CFU (25–32% reduction). Measurements of the relative ATP content using the BacTiter^Glo^ microbial cell viability assay showed that AEA alone or in combination with MET reduced the ATP content in comparison with control and MET-treated bacteria (Figure 1E). Of note, there was a transient increase in the ATP content in the samples treated with both AEA and MET (Figure 1F), which might be due to a release of ATP to the surroundings upon cell death that accounts for the reduced CFU (Figure 1D) and/or an inhibition of ATPase that will be discussed below. The ATP content after 24 h for the combined treatment was still low, similar to that observed after an 8 h incubation. Altogether, these assays show that the combined treatment of AEA and MET was bactericidal to MDRSA CI-M during the first 8 h of incubation, while being bacteriostatic thereafter.

### 2.2. Anandamide Increases the Intracellular Level of Bocillin-FL

In our previous study, AEA was shown to prevent the extrusion of the efflux pump substrates DAPI and ethidium bromide, as well as the antibiotic norfloxacin [49], known to be pumped out by the NorA-C efflux pumps [52]. A major question is whether a similar mechanism is involved in the sensitization of MDRSA to β-lactams. Resistance to β-lactam antibiotics is usually caused by penicillinases that degrade the β-lactam antibiotics as well as the expression of variant penicillin-binding proteins such as PBP2a that shows low affinity to penicillin [13,53]. In addition, the ATP-dependent AbcA efflux pump can confer β-lactam resistance by extruding the drug [54]. We found that the *abcA* mRNA was present in MDRSA CI-M, but its gene expression was not altered after a 2 h-exposure to 50 µg/mL AEA (Appendix A). Previous studies [49] showed that AEA even increased the expression of *mecA* that encodes for the low penicillin affinity PBP2a that was reconfirmed here (Appendix A). Thus, the AEA-mediated sensitization to MET is likely to occur at a post-transcriptional level. We therefore studied whether AEA could increase the intracellular levels of β-lactams. To this end, we exposed both methicillin-sensitive (MSSA 25923) and methicillin-resistant *S. aureus* (MDRSA CI-M) to the fluorescent penicillin analog Bocillin-FL in the absence or presence of AEA, and the intensity of the green fluorescence was studied by flow cytometry. Indeed, we observed that AEA increased the intracellular Bocillin-FL concentration in both *S. aureus* strains (Figure 2A,B), suggesting that AEA leads to intracellular accumulation of β-lactams. The accumulation of Bocillin-FL in the presence of AEA was more profound in the MSSA 25923 strain than in the MDRSA CI-M strain (Figure 2A versus Figure 2B). 

We next studied the Bocillin-FL staining of the MDRSA CI-M bacteria that were incubated for 2 h in the absence or presence of 50 µg/mL AEA by spinning disk confocal microscopy. Clear cell wall and septum staining of control bacteria were observed (Figure 2C), indicating that the bacteria express penicillin-binding proteins in the cell wall that can bind Bocillin-FL. However, in AEA-treated bacteria, a more diffuse, but stronger Bocillin-FL staining was observed (Figure 2D), suggesting an increased intracellular level of Bocillin-FL and different bacterial distribution of penicillin-binding proteins following AEA treatment.

We next extracted proteins from control and AEA-treated MDRSA CI-M that were incubated with Bocillin-FL. The soluble fraction was separated on an SDS-PAGE gel under reducing conditions to reveal free Bocillin-FL (Figure 3A), while the membrane fraction was separated on an SDS-PAGE gel under non-reducing conditions to reveal protein-bound Bocillin-FL (Figure 3B). Under reducing conditions, Bocillin-FL was separated from the proteins and ran at the front of the gel. We could observe a higher amount of Bocillin-FL in the AEA-treated MDRSA CI-M and AEA-treated MSSA 25923 than in the respective control bacteria (Figure 3A), supporting our above-described observations that AEA increases the intracellular concentrations of Bocillin-FL (Figure 2). Under non-reducing conditions, Bocillin-FL binding to various bacterial membrane proteins could be observed (Figure 3B). There were more proteins binding Bocillin-FL in the AEA-treated bacteria than in the control bacteria, which was more pronounced in MSSA 25923 than in MDRSA CI-M (Figure 3B). Altogether, these data suggest that AEA causes an accumulation of β-lactams within the bacteria as well as enabling them to bind to PBPs.

### 2.3. Anandamide Causes an Intracellular Accumulation of Its Own

As AEA is a fatty acid derivative of arachidonic acid; it was anticipated that AEA would be mainly found in the bacterial membrane. To test this possibility, we treated MDRSA CI-M with 5 µg/mL of the fluorescent AEA derivate SKM 4-45-1 in the absence or presence of 50 µg/mL AEA for 2 and 4 h, and the green fluorescence was visualized by spinning disk confocal microscopy. SKM 4-45-1 does not emit fluorescence outside the cell, but is converted into a fluorescent compound by intracellular esterases [55]. The fluorescent AEA caused weak staining of the bacteria, which was enhanced in a time-dependent manner in the presence of unstained AEA (Figure 4A), suggesting that AEA prevents the export of the fluorescent AEA. The fluorescent AEA stained the whole bacteria and not only the membrane, indicating that its localization is not restricted to the membrane (Figure 4A). The uptake of fluorescent AEA was also measured in a time-course study in a fluorescent plate reader, and the fluorescence intensity was increased over time (Figure 4B), suggesting an increase in its intracellular concentration with time.

### 2.4. Anandamide Prevents the Secretion of Exopolysaccharides (EPS)

Since we observed that AEA prevents the extrusion of drugs, it was interesting to study whether it also affects the secretion of EPS, which is important for biofilm formation. To this end, MRDSA CI-M was exposed to 12.5 or 50 µg/mL AEA for 1 h, and then a 10 µL bacterial culture of each triplicate was inoculated on Congo Red agar plates containing 5% sorbitol or plain Congo Red agar plates. After an overnight incubation, black staining indicative of exopolysaccharide production was observed around colonies of control bacteria grown on the sorbitol plates (Figure 5A), while no black staining was observed on the Congo red plates without sorbitol (Figure 5B). After treatment with 12.5 µg/mL AEA, there was a reduction in black staining on the Congo Red plates containing sorbitol, and no black staining was observed in the bacterial culture that was treated with 50 µg/mL AEA for 1 h (Figure 5A). Similar data were observed after a 4 h incubation with AEA (data not shown). These data suggest that AEA inhibits the secretion of EPS which may be one mechanism contributing to its anti-biofilm activity [49,51]. Previous studies showed that AEA reduced the expression of the *icaA* and *icaB* genes encoding polysaccharide intercellular adhesin (PIA), by 50–75% [49]. The reduced expression of the *ica* genes might in part explain the reduced secretion of EPS by AEA.

### 2.5. Anandamide Increases the Extracellular DNA (eDNA) Level

The release of eDNA from a subpopulation of bacteria that have undergone lysis is important for biofilm formation [56,57]. The eDNA binds to the cell surface of the other bacteria, keeping them clustered together [56]. Since AEA prevents biofilm formation [49,51] and we observed reduced EPS secretion (Figure 5A), it prompted us to study the effect of AEA on eDNA production. We first seeded control and AEA (25 μg/mL or 50 μg/mL, 1 h)-treated MDRSA CI-M on TSA plates that were supplemented with methyl green that stains DNA. Surprisingly, the presence of 0.02% methyl green in TSA prevented the growth of the AEA-treated bacteria (Appendix A). The same AEA-treated bacteria grew normally on TSA without methyl green (Appendix A), suggesting that AEA treatment makes the bacteria sensitive to methyl green. Notably, the tiny colony formed by the 1 h 25 μg/mL AEA-treated bacteria in the presence of methyl green appeared in blue color (Appendix A), suggesting the presence of eDNA. The colony of the control bacteria was not colored blue, and a small halo could be seen around the colony (Appendix A). 

To further analyze the effect of AEA on eDNA production, we incubated MDRSA CI-M in the absence or presence of increasing concentrations (1.56–50 μg/mL) of AEA for 2 h, followed by DNA staining using three different DNA stains: the cell-impermeable TOTO-1 iodide that only stains eDNA, SYTO 9 that freely diffuses into the bacteria and stains the bacterial DNA, and PI that stains eDNA and can only enter bacteria with comprised membranes [58,59]. A basic TOTO-1 staining was observed for control bacteria (Figure 5C), suggesting the already presence of eDNA on the surface of these bacteria. Upon treatment with 25 and 50 μg/mL AEA, two peaks were observed on flow cytometry (Figure 5C and Appendix A). The major one showed similar fluorescence intensity as the control bacteria, while the second peak of stronger fluorescence intensity expanded with increasing concentrations of AEA, reaching up to 40% of the culture treated with 50 μg/mL AEA (Figure 5C,F). When looking at the dot plots of TOTO-1 staining versus forward scatter (FSC) (Appendix A), the cell population with higher TOTO-1 fluorescence intensity showed even distribution along the FSC axis that was similar to the cell population with low TOTO-1 fluorescence intensity, indicating that the bacteria were alive, except for a minor cell population with an FSC lower than 80 that likely represents shrunk dead cells. These findings suggest that AEA increases the production of eDNA, likely through cell death of a minor subpopulation of the bacteria.

Strikingly different fluorescent intensity distribution was observed between control and AEA-treated bacteria when staining with SYTO 9 and PI (Figure 5D,E). In SYTO 9-stained control bacteria, the main peak was followed by a tail to the right (Figure 5D), while in the SYTO 9-stained AEA-treated bacteria, there were two peaks, with the one to the right dominating (Figure 5D). A similar pattern was observed for 12.5, 25, and 50 μg/mL AEA (Figure 5D,F). Since SYTO 9 freely penetrates the bacteria and the fluorescent intensity reflects the relative amount of nucleic acids in each cell, the appearance of the two peaks suggests that AEA causes an increase in the DNA content. This confirms our previous study [49] showing that AEA prevents cell division just before daughter cell separation. Concerning the PI staining, a major peak with a tail to the right was observed for the control bacteria (Figure 5E), again pointing to a certain basal level of eDNA bound to the bacteria surface. AEA treatment led to the appearance of a cell population with a higher PI fluorescent intensity that was dose-dependent, reaching up to 80% of the culture with 50 μg/mL AEA (Figure 5E,F). Altogether, these data show that AEA increases the release of eDNA. 

### 2.6. AEA Does Not Alter the Surface Binding of WGA, ConA, or Dextran to MDRSA CI-M

Next, we wanted to know if AEA affects the general surface carbohydrate composition. To this end, we incubated control and AEA (50 µg/mL, 2 h)-treated MDRSA CI-M with the fluorescence-conjugated lectin wheat germ agglutinin (WGA) that binds to sialic acid and N-acetylglucosaminyl residues, or the fluorescence-conjugated lectin Concanavalin A (ConA) that binds to α-D-mannosyl residues. In addition, the control and treated bacteria were exposed to fluorescence-conjugated anionic dextran (10,000 Dalton) that binds to positive charges on the bacterial surface. We observed that WGA had a high affinity to the surface of MDRSA CI-M that was not changed by AEA (Appendix A). ConA bound weakly to MDRSA CI-M, and this binding was also unaffected by AEA (Appendix A). Anionic dextran showed moderate binding to MDRSA CI-M, with no changes upon AEA treatment (Appendix A). These data suggest that AEA does not have a major impact on the carbohydrate composition of the cell wall components and the overall net charge of the cell wall is retained.

### 2.7. AEA Reduces the Cell Wall Teichoic Acid (WTA) Content 

Alteration in cell wall teichoic acid (WTA) content and composition has been shown to affect both biofilm formation and the sensitivity of the bacteria to antibiotics and other stress stimuli [38,39,40,42,60,61,62,63,64]. Since WTA is initially synthesized on an undecaprenyl phosphate (bactoprenol) lipid carrier inside the bacterial cell before being transported to the cell surface where they are covalently linked to proteoglycans [41], it was compelling to study whether AEA, which is a fatty acid derivative, could interfere with WTA synthesis. To this end, the WTA was isolated from the cell wall of control and AEA (50 μg/mL, 2 h)-treated MDRSA bacteria and analyzed on a 20% native polyacrylamide gel and visualized by alcian blue staining followed by silver staining. This analysis showed that AEA significantly reduced the WTA content of the cell wall (Figure 6A). In parallel, the SDS extracts prepared from the same samples during the process of WTA isolation were separated on a 10% SDS-PAGE gel that served as loading controls (Figure 6B). Of note, the size spectrum of the WTA was not changed, indicating that AEA did not affect its glycosylation (Figure 6A).

The reduced WTA cell wall content following AEA treatment could be due to a direct reduction in WTA biosynthesis, or indirectly, through modulation of the expression of genes involved in WTA synthesis (see Appendix A for a brief description of the function of the tested *tar* genes). Of the six WTA-related genes tested, an exceptionally strong increase in *tarG* expression (6.6 ± 0.7 fold) was observed (Appendix A), with a less profound, but still significant, increase in the expression of *tarA*, *tarH*, *tarM*, and *tarS* (1.5–2.2 fold) (Appendix A). TarG together with TarH forms the ABC transporter that transfers the wall teichoic acid across the membrane [41]. No significant effect was observed on *tarO* expression (Appendix A) which is involved in the initial step of cell wall teichoic acid synthesis [41]. Altogether, these data indicate that AEA prevents the formation of WTA. The changes in WTA-related genes might reflect a compensatory mechanism to overcome the reduced formation of WTA in the presence of AEA.

### 2.8. AEA Alters the Membrane Fluidity and Composition of MDRSA CI-M

In our previous study, we showed that AEA caused alterations in the membrane structure [49]. Since the membrane transport systems are affected by membrane fluidity and membrane lipid composition [65], it was of interest to study the effect of AEA on membrane fluidity. A commonly used agent for studying membrane fluidity is Laurdan, which is a fluorescence probe that intercalates into the membrane bilayer and displays an emission wavelength shift depending on the number of water molecules in the membrane [66,67]. Laurdan is an amphipathic molecule with a long aliphatic tail that integrates into the membrane and a fluorescent moiety exposed to the aqueous phase [68]. Following a 5 h treatment with various concentrations of AEA or in its absence, the MDRSA CI-M was loaded with Laurdan, and the fluorescence spectrum from 400 nm to 600 nm was measured upon excitation at 350 nm. We observed that treatment of MDRSA CI-M with low concentrations of AEA (1.56 and 3.125 µg/mL) caused an increase in the amount of Laurdan incorporated into the membrane, while at higher concentrations there was a dose-dependent drop in Laurdan incorporation (Figure 7A), suggesting that AEA treatment causes alterations in the membrane structure making it less available for Laurdan incorporation. The generalized polarization (GP) value that quantifies the degree of dipolar relaxation of the membrane was more or less the same within the concentration range of 1.56–25 µg/mL AEA, but started to drop at 50 µg/mL with a strong reduction at 100 µg/mL AEA (Figure 7B), indicative of a more rigid membrane.

Since cardiolipin may affect membrane microdomains, membrane structure, and antibiotic sensitivity [69,70,71], we analyzed its content by two different approaches. In the first approach, we used the 10-N-nonyl-acridine orange (NAO) which is known for its ability to bind to cardiolipin, and upon cardiolipin binding the dye emission shifts from 525 nm (green fluorescence) to 640 nm (red fluorescence) [72]. NAO labeling of MDRSA CI-M bacteria that were treated with 25 or 50 μg/mL AEA for 2 h, led to rapid spontaneous lysis of the bacteria. Therefore, the analysis was performed on 12.5 µg/mL AEA-treated bacteria. Flow cytometry analysis showed that the AEA (12.5 μg/mL, 2 h)-treated MDRSA CI-M emitted a 2-fold increase in green fluorescence intensity in comparison with control bacteria (Figure 7C), while the red fluorescence intensity increased 5.4-fold (Figure 7D). The increase in green fluorescence observed in the 12.5 μg/mL AEA-treated bacteria could be due to the drug retention effect of AEA as discussed above. When correcting for green fluorescence, the relative ratio of red/green fluorescence of AEA-treated bacteria was 2.5-fold higher than in control bacteria, suggesting an increased cardiolipin content. To study this issue further, the membrane lipid content of control and AEA-treated MDRSA CI-M was compared by running a two-dimensional thin-layer chromatography (TLC) (Figure 7E,F). The TLC analysis shows that AEA treatment caused a relative increase in the cardiolipin spot when compared with control bacteria and using the phosphatidylglycerol spot as reference (Figure 7F versus Figure 7E). Moreover, the cardiolipin spot of the AEA-treated bacteria showed a shift, suggesting that AEA affects the composition of the cardiolipin.

### 2.9. Anandamide Inhibits the Membrane ATPase Activity, but Not the Membrane GTPase Activity

Since many of the active membrane transport systems require energy from ATP, either directly or indirectly [73], it was important to study whether AEA affects the membrane ATPase activity. For this purpose, membranes from MDRSA CI-M that were exposed to 12.5 or 50 µg/mL AEA for 30 min or control bacteria were isolated, and their ability to hydrolyze ATP or GTP was measured in a kinetic study. Both concentrations of AEA inhibited the general membrane ATPase activity by 47–53% (Figure 8A), while barely having any effect on the general membrane GTPase activity (Figure 8B).

### 2.10. The Divisome Z-Ring Is Formed in the Presence of AEA

Previous studies showed that AEA inhibits bacterial growth of MDRSA CI-M by preventing the cell division step of daughter cell separation [49]. Bacterial cell division involves the polymerization of FtsZ into a midcell ring-like structure termed Z-ring that acts as a scaffold for the recruitment of additional cell division proteins including EzrA, SepF, FtsA, GpsB, ZapA, DivIVA, and DivIBC that together form the divisome [74,75,76]. To test whether AEA affects FtsZ Z-ring formation, an *S. aureus* strain overexpressing FtsZ-GFP was exposed to 50 µg/mL AEA for 2 h and the green fluorescence was visualized by spinning disk confocal microscopy. The FtsZ Z-ring was observed in bacteria exposed to AEA (Figure 9B and Appendix A), suggesting that AEA does not interfere with the formation of the Z-ring that requires intact FtsZ GTPase activity [77,78]. It was clearly seen that there were relatively more bacteria with FtsZ Z-ring in the AEA-treated bacteria than in control bacteria (Figure 9A,B and Appendix A), suggesting that the growth inhibition occurs at a step after Z-ring formation. Besides being seen in the Z-ring of dividing bacteria, FtsZ-GFP was observed in the cell membrane (Figure 9A,B and Appendix A). 

DivIVA that has been shown to interact with FtsZ [75] exhibited similar dispersed intracellular staining in control and AEA-treated MDRSA CI-M (Figure 9C,D and Appendix A). EzrA that interacts with FtsA in the divisome [75] was found both in the membrane and in the Z-ring of AEA-treated MDRSA CI-M (Figure 9E,F and Appendix A), suggesting that AEA does not interfere with the localization of EzrA to the divisome. The chaperone DnaK that interacts with FtsZ, EzrA, and DivIVA [75] showed similar intracellular staining in control and AEA-treated bacteria (Appendix A). These data suggest that AEA does not interfere with the formation of the divisome.

To study whether overexpression of FtsZ-GFP, EzrA-GFP, DivIVA-GFP, or DnaK could antagonize the growth-inhibitory effect of AEA, these proteins were induced in the bacteria before exposure to AEA followed by a kinetic growth assay. The overexpression of any of the four proteins did not prevent the AEA-mediated cell growth arrest (Appendix A), further supporting the hypothesis that AEA prevents daughter cell separation by acting on a step after Z-ring formation.

Eswara et al. [76] observed that lack of the gene *gpsB* halted cell division and caused cell lysis, whereas overexpression of GpsB led to the formation of enlarged cells due to too early activation of FtsZ. We therefore studied the effect of AEA treatment on *gpsB* gene expression and found that AEA indeed reduced *gpsB* expression by 53 ± 5% after a 2 h exposure (Appendix A) that might contribute to the growth inhibitory effect of AEA. 

### 2.11. AEA Induces Spontaneous Cell Lysis despite the Reduced Activity of Autolysins

Autolysins or murein hydrolases are autolytic enzymes that hydrolyze the mucopeptide polymers in the bacterial cell wall [79]. These enzymes are involved in both bacterial lysis and cell division. A timely activation of autolysins is required for proper remodeling of the cell wall during cell division and for daughter cell separation [80,81,82,83]. Since AEA interrupts cell division at the stage of daughter cell separation as discussed above, it was interesting to study the effect of AEA on autolysis. To this end, we applied the Triton X-100-induced autolysis assay [84] where the control and 2 h AEA-treated bacteria were washed twice in sterile water followed by exposure to 50 mM Tris pH 7.6 in the absence or presence of 0.05% Triton X-100. In five independent experiments, we repeatedly observed that the AEA-treated bacteria showed rapid initial lysis in the absence of Triton-X-100 (Figure 10A). While Triton X-100 induced autolysis in control MDRSA CI-M bacteria, it did not significantly increase the autolysis of AEA-treated bacteria, at least within the 6 h of analysis (Figure 10A). The reduced sensitivity of AEA-treated bacteria to Triton X-100 may be due to the AEA-induced membrane alterations and/or changes in the expression of autolysins. 

Banerjee et al. [49] observed a reduction in gene expression of the autolysins *sle1*, *lytA*, *lytM*, and *lytN* in bacteria exposed to AEA for 4 h. Here we showed that after a shorter incubation time of 2 h with 50 μg/mL AEA, the *sle1* level was not significantly altered and the *atl1* autolysin gene expression was slightly reduced (Appendix A). We further observed that AEA increased the gene expression of both the holin *cidA* and the anti-holin *lrgA* (Appendix A) that respectively increases and reduces autolysin activities [85,86]. The small non-coding RNA *sprX* that regulates autolysin expression [87] was upregulated 3-fold in AEA-treated bacteria (Appendix A), while the autolysin regulator *walR* was only modestly increased (1.5-fold) (Appendix A). The two-component system regulator *lytR* that promotes the expression of the anti-holin *lrgA* [88] was also modestly upregulated (1.8-fold) by AEA (Appendix A). The contrasting effects of AEA on autolysin regulatory factors might be compensatory mechanisms to adapt to the AEA-induced stress responses.

Due to the inconclusiveness of the gene expression studies, we performed zymography to study the autolysin activities from AEA and/or MET-treated MDRSA CI-M compared with control bacteria. The identity of the observed autolysin bands was interpreted according to the study of Vaz and Filipe [89]. A truncated form of Atl lacking the GL domain was found to be similarly expressed in control and AEA and/or MET-treated bacteria, while the mature amidase (AM) of Atl was significantly reduced in AEA-treated cells (Figure 10B). Of special importance is the significant downregulation in the activity of the autolysin Sle1 by AEA or MET alone, with an even stronger reduction when applied in combination (Figure 10B). Since Sle1 and AtlA are involved in daughter cell separation [90,91], the reduced expression of these autolysins can, at least partially, contribute to the growth inhibitory effect of AEA.

**Figure 10 ijms-23-07798-f010:**
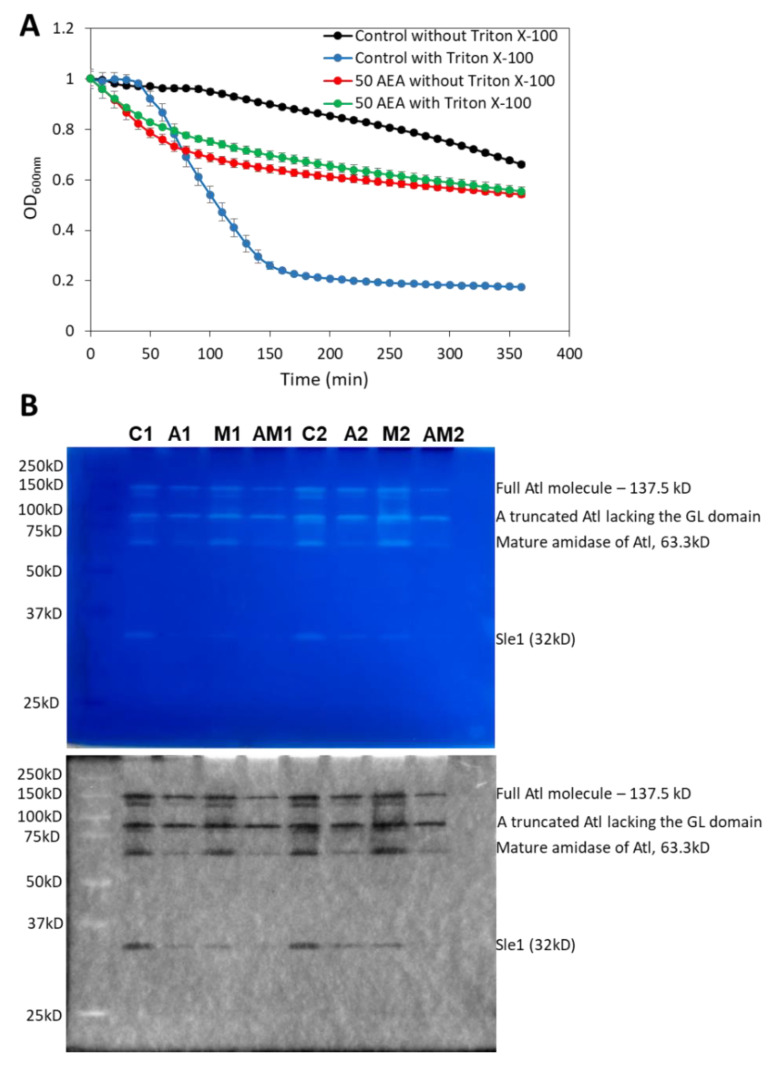
**AEA treatment leads to increased spontaneous lysis, but reduced cell wall autolysin Sle-1 and mature Atl1 activities.** (**A**). Kinetic autolysis assay of MDRSA CI-M that has been incubated in the absence or presence of 50 μg/mL AEA for 2 h, followed by two washes in sterile water, and incubation in 50 mM Tris HCl pH 7.6 in the absence or presence of 0.05% Triton X-100 at 37 °C. (**B**). Zymogram of autolysin extracts of MDRSA CI-M that has been incubated in the absence or presence of 50 μg/mL AEA and/or 50 μg/mL MET for 2 h using *S. aureus* cell wall as substrate in a 10% SDS-PAGE. Duplicate samples are presented (Series 1 and 2). C = Control; A = AEA; M = Met; AM = AEA + MET. The upper panel is the Coomassie blue staining as captured by a Samsung smartphone camera, while the lower panel is the inverted gel image as captured by the Bio-Rad ChemiDoc XRS+ imaging system. The identities of the autolysin bands were determined according to Vaz and Filipe [89].

Since Sle1 is degraded by the ClpXP protease [90], the effect of AEA on *clpP* mRNA level was studied. AEA was found to increase *clpP* gene expression 2.5-fold (Appendix A), which might contribute to the observed reduction in the autolytic activity of Sle1. We then analyzed the protease activities of control and AEA (50 μg/mL, 2 h)-treated MDRSA CI-M by gelatin and casein zymography. The gelatin and casein zymograms showed the appearance of a high-molecular-weight proteolytic activity of >>250 kD following AEA treatment that was not observed in control samples (Appendix A). This protease might represent the ClpXP which is a high-molecular-weight multimeric complex [92]. A gelatinolytic activity of a ~38 D protease was observed in both control and treated samples, which was weaker in the AEA-treated samples (Appendix A). A caseinolytic activity of a ~22 kD protease was observed in AEA and MET treated samples, which was barely seen in the control samples (Appendix A). The ability of the MDRSA to secrete gelatinolytic proteases was not disturbed by AEA (Appendix A). 

We further observed that AEA strongly reduced the gene expression of the redox sensitive transcriptional factor *spx* by around 80% (Appendix A) that confers resistance to oxacillin by controlling the expression of TrfA that acts as an adaptor molecule assisting ClpXP-mediated degradation of the MazE antitoxin [93,94]. The repression of spx by AEA might be important for overcoming *spx*-induced antibiotic resistance. 

## 3. Discussion

Methicillin-resistant and multidrug-resistant *S. aureus* (MRSA/MDRSA) are major threats of nosocomial infections, and new strategies are required to combat these infections, which are often complicated by biofilm formation on biotic and abiotic surfaces. Many approaches have been undertaken to overcome the drug resistance of these bacteria. The first attempt was to use β-lactamase inhibitors to neutralize the enzymes responsible for the degradation of β-lactams [95]. Next, methicillin was developed that resists degradation by β-lactamase [13]. However, within a short time after introducing methicillin to the drug market, a different kind of β-lactam resistance appeared, namely the expression of a low affinity penicillin-binding protein PBP2a that could complete cell wall synthesis in the presence of methicillin [13]. This led to the development of a new generation of β-lactams such as the cephalosporin ceftaroline, which can bind to the allosteric site of PBP2a, resulting in a conformation change that enables the binding of another drug molecule to the active site with consequent inhibition of its enzymatic activity [35]. Another prevailing drug resistance mechanism is the overexpression of efflux pumps that promote the extrusion of antibiotics, resulting in low intracellular drug concentrations that are insufficient for the anti-microbial action [14]. In this respect, efflux pump inhibitors have been developed in an attempt to sensitize the drug resistant bacteria to antibiotics [96]. This approach has the advantage that antibiotics already existing in the drug market can be reused. The application of efflux pump inhibitors revealed a connection between efflux pumps and biofilm formation [97,98,99]. By inhibiting efflux pumps, the biofilm formation ability was strongly reduced [97,98,99].

We previously observed that the endocannabinoid Anandamide (AEA) could sensitize both MRSA and MDRSA to various antibiotics and prevent biofilm formation ([49,50,51] and reviewed recently by Sionov and Steinberg [100]). One of its action mechanisms was found to be the inhibition of drug efflux [49], which might explain the ability of AEA to sensitize the bacteria to various classes of antibiotics [49,50]. Additionally, AEA was shown to cause an immediate growth arrest that was attributed to a prevention of the late cell division stage of daughter cell separation [49]. Moreover, AEA caused an immediate membrane depolarization as well as alterations in the membrane structure [49]. Based on these findings, we were interested in understanding in more depth the underlying biochemical changes caused by AEA.

At the first step, it was important to study whether the combined treatment of AEA with methicillin (MET) had a bacteriostatic or bactericidal effect. This was achieved by following changes in CFUs in a kinetic study. During the first 8 h of incubation, there was a 10-fold reduction in CFU, indicating that the combined treatment had a bactericidal effect. It is worth mentioning that the growth inhibition caused by AEA did not interfere with the antibacterial activity of penicillin antibiotics, which have previously been shown to require cell division in order to act [24,101]. As a single drug, cell division is required for the anti-bacterial β-lactam activity since the target is cell wall synthesis. However, when combined with drugs such as FtsZ inhibitors [102,103,104] or, in our case, AEA, the anti-bacterial activity of β-lactam antibiotics was restored even when the growth was stalled. This phenomenon might be in part explained by the requirement for coordinated cell division for bacterial survival [75,76,82]. 

Since AEA causes growth arrest and FtsZ is required for cell division by forming a Z-ring at the septum involving interaction with several other proteins including PBP2 [36,74,76,105], it was compelling to study the effect of AEA on FtsZ Z-ring formation. FtsZ monomers are polymerized into FtsZ filaments in the presence of GTP, which forms FtsZ bundles in the presence of GpsB [76]. The GTPase activity of FtsZ is then activated, resulting in FtsZ disassembly [76]. Inhibition of GTPase activity or depletion of GpsB leads to cell growth arrest [36,76]. We observed that AEA did not interfere with FtsZ Z-ring formation, and this cell division protein localized to the septum in the presence of AEA. In accordance with our previous study [49], there was a significant increase in the number of bacteria with a complete septum. This observation together with similar membrane GTPase activity in the absence or presence of AEA suggests that AEA prevents cell division at a step after Z-ring formation. Gene expression studies showed a reduction in *gpsB*, which might contribute to the inhibition of cell division. 

Outstanding was the strong reduction in the autolytic activities of the autolysins Sle1 and mature Atl1 known to be crucial for proper daughter cell separation [90,91]. This effect seems to be the major contributor to the cell division arrest at the late stage of cell division by AEA. Sle1 has been shown to be essential for β-lactam resistance in community-acquired MRSA [90], such that reduction in its expression by AEA may contribute to the sensitization to β-lactam antibiotics. Methicillin alone also reduced the Sle1 level, which confirms the finding of Thalsø-Madsen et al. [90] showing that the β-lactam oxacillin decreases the Sle1 level with a consequent cell separation defect. Importantly, the combined treatment of AEA with MET caused a stronger reduction in Sle1 autolytic activity than either compound alone. Additionally, AEA was found to increase the gene expression of the *clpP* protease that is involved in the degradation of Sle1 [90] and the transcription factor Spx [106]. Inactivation of the ClpXP protease activity has been shown to augment β-lactam resistance in an MRSA strain [107], while overexpression of Spx imposed oxacillin resistance [94,108]. Notably, AEA caused an 80% reduction in *spx* gene expression, which should mitigate the resistance phenotype. Both gelatin and casein zymograms show the appearance of a high molecular weight (>>250 kD) proteolytic activity following AEA treatment, which might represent the multimeric ClpXP complex [109]. Altogether, these data show that AEA affects both protease and autolysin activities, resulting in an imbalance that has an impact on cell viability, cell division, antibiotic susceptibility, and biofilm formation. 

The intracellular drug concentration is an important factor for treatment success. Previous studies showed that AEA prevents drug efflux of norfloxacin as well as the efflux pump substrates ethidium bromide and DAPI [49]. Since β-lactams have been shown to be pumped out by the AbcA efflux pump [54], it was of interest to study the effect of AEA on the intracellular concentration in β-lactam using the fluorescent Bocillin-FL. We observed that AEA increased the intracellular concentration of Bocillin-FL that was more prominent in the MSSA strain than in the MDRSA strain, and the Bocillin-FL binding to various membrane proteins was increased. This suggests that AEA not only prevents the efflux of β-lactam antibiotics, but also increases the binding of the drugs to their targets. Confocal microscopy shows a different distribution of the Bocillin-FL in the AEA-treated bacteria than in the control bacteria. This goes along with previous observations that AEA causes altered membrane structures [49]. The altered membrane structures might also explain the dose-dependent reduction in Laurdan membrane incorporation into the AEA-treated bacteria. Interestingly, AEA caused an increase in the relative cardiolipin membrane content.

A major question is how AEA prevents drug efflux. One mechanism might be the altered membrane structure that affects efflux pump activity and solute transport. Another possibility is an effect on membrane ATPase activity. ATP turnover is important for many of the basic bacterial processes including maintaining the membrane potential, membrane transport, metabolism, and cell division [73,110,111]. Indeed, AEA was found to inhibit membrane ATPase activity. The reduced membrane ATPase activity caused by AEA may explain many of the effects observed for this compound including membrane depolarization, inhibition of drug efflux, inhibition of cell wall teichoic acid content, and impaired EPS production, with implications for drug sensitization and impaired biofilm formation. 

Membrane depolarization has been shown to affect the membrane distribution of cell division proteins [111], increase the susceptibility to antibiotics [73], and impede biofilm formation [112]. The LytSR regulatory system induces the transcription of the *lrgAB* operon in response to membrane depolarization [113], and the antiholin-like protein LrgA in turn inhibits autolysis [114]. In this context, it is noteworthy that AEA increased the gene expression of *lytR* and *lrgA* as well as that of the holin *cidA*, whose expression is regulated by the LysR-like transcription factor CidR and the catabolite control protein A (CcpA) [115]. CidA acts antagonistically with LrgA by promoting autolysis and eDNA release [85,86]. Simultaneous, AEA induced a 3-fold increase of the small non-coding *sprX* that positively regulates the expression of the autolysin regulator WalR, which, in turn, modulates the expression of the autolysins *isaA* and *lytM* [87,116,117]. SprX has been shown to sensitize MRSA to antibiotics by negatively regulating the translation of the transcription factor *spoVG*, which is a master regulator of capsule production, virulence, and cell-wall metabolism, and required for methicillin and glycopeptide resistance [118,119,120]. SpoVG negatively regulates the expression of the autolysin *lytN*, while it positively regulates the expression of the *lytSR* two-component system and *femAB* that catalyze the formation of the pentaglycine interpeptide bridge of the peptidoglycan [120]. Thus, contrasting forces are acting simultaneously on autolysins. 

Despite reduced autolysin activity, AEA treatment resulted in an increased release of eDNA, which is usually produced upon cell lysis of a bacterial subpopulation [121]. This finding suggests that a certain extent of cell lysis took place upon AEA exposure. Indeed, flow cytometry analysis of TOTO-1-stained AEA-treated bacteria showed a minor cell population with high TOTO-1 fluorescence intensity and low forward scatter (FCS), which might represent dying bacteria responsible for the eDNA release. Additionally, spontaneous cell lysis of AEA-treated MDRSA was observed in the autolysis assay, despite resistance to Triton X-100-induced autolysis. Liu et al. [122] observed that an *S. aureus clpP* mutant that shows elevated Sle1 levels, formed better biofilms, and was more susceptible to Triton X-100 induced autolysis. Taken together, these data suggest that the increased *clpP* expression observed upon AEA treatment could contribute to the AEA-induced resistance to Triton X-100-induced autolysis as well as the AEA-mediated reduction in biofilm formation. It is worth noting that AEA not only increased the susceptibility of drug-resistant *S. aureus* to various antibiotics [49,50], but also to the cationic methyl green and the cardiolipin-binding dye NAO (described in this paper). The general sensitization of AEA to multiple compounds might be due to an increased proneness to cell lysis. 

The release of eDNA is a mechanism evolved by the bacteria to increase bacterial co-adherence and promote biofilm formation. However, AEA treatment ultimately prevents biofilm formation [49,50,51], indicating that the production of eDNA per se is not sufficient for biofilm formation, and requires additional factors. Using Congo Red as an indicator for EPS secretion, we demonstrated that even a short time incubation (1 h) with AEA was sufficient to prevent EPS secretion for the next 24 h, despite the recovery of the bacteria on the agar plates. The prevention of EPS secretion goes along with the inhibition of drug efflux and the inhibition of membrane ATPase activity required for these processes.

Another process that depends on ATP is the membrane transport of the phosphate-rich, sugar-based polymer cell wall teichoic acid (WTA) [41] that affects several bacterial cell processes including survival, vitality, cell division, biofilm formation, virulence, and antibiotic resistance [39,63,123,124]. The WTA biosynthesis is a multiple step process that is initiated by TagO that catalyzes the transfer of GlcNAc-phosphate to the undecaprenyl (bactoprenol) phosphate scaffold followed by the addition of ManNAc by TarA, two glycerol-phosphate units by TarF, and 20–40 ribitol-phosphate repeats by TarL [41]. TarM and TarS add α-linked and β-linked N-acetylglucosamine, respectively, to the polyribitol chain of the growing WTA [41]. The mature WTA is then transported through the membrane by the ABC-transporter TarGH which is driven by ATP hydrolysis, followed by covalent attachment to the proteoglycan cell wall by TagTUV [41]. AEA strongly reduced the WTA content of the cell wall without affecting the molecular sizes of WTA, suggesting that AEA inhibits the ATP-dependent membrane transport of WTA similar to the action of targocil [124]. A hint for this mechanism is the extreme upregulation of *tarG* expression in response to AEA, which seems to be a compensation mechanism for its blockage. Campbell et al. [39] previously demonstrated that blocking WTA synthesis at the TarO enzymatic step by tunicamycin reduced the MIC of oxacillin by 128-fold in some MRSA strains, suggesting a central role for WTA in conferring antibiotic resistance. Tiwari et al. [84] observed that untranslocated WTA molecules sequestered Atl at the membrane, resulting in reduced autolysin level in the cell wall with consequently decreased autolysis. This scenario resembles the observations of AEA-treated MDRSA showing reduced WTA content in the cell wall and reduced cell wall-associated autolysins Sle1 and Atl1. 

## 4. Materials and Methods

### 4.1. Materials

Anandamide (AEA) (>97.0% purity) was purchased from Sigma-Aldrich (St. Louis, MO, USA), while methicillin sodium salt was obtained from Cayman Chemical (Ann Arbor, MI, USA). Stock solutions of AEA and methicillin were prepared in absolute ethanol at 10 mg/mL, and respective ethanol dilutions were used as controls.

### 4.2. Bacterial Strains and Culture Conditions

The major microbial strains used in this study were MSSA *S. aureus* ATCC 25923 and MDRSA clinical isolate M (CI-M). MDRSA CI-M is defined as MDRSA as it is insensitive to methicillin, norfloxacin, gentamicin, and erythromycin, and it expresses multiple efflux pumps as determined by real-time PCR [49]. In addition, we used the following *S. aureus* strains kindly provided by Prof. Elizabeth Harry (The iThree Institute, Faculty of Science, University of Technology Sydney, Australia) [75]: LH607 (8325-4 *spa*::*tet* (*tet*^R^)), LH607 with inducible FtsZ-GFP (SA103; pLOW *ftsZ*-*gfp*, pGL485 (*erm*^R^, *cat*^R^, *tet*^R^)), LH607 with inducible GFP (SA112; pLOW GFP, pGL485 (*tet*^R^, *erm*^R^ *cat*^R^)), LH607 expressing EzrA-GFP (SA353; *ezrA*::*ezrA*-*gfp*, pGL485 (*erm*^R^ *cat*^R^)), LH607 expressing inducible DivIA-GFP (SA356; P*_divIVA_ divIVA*-*gfp*::P*_spac_ divIVA*, pGL485 (*erm*^R^ *cat*^R^ *tet*^R^)), and RN4220 with inducible DnaK-GFP (SA307; pLOW *dnaK*-*msgfp*, pGL485 (*erm*^R^ *cat*^R^)). pLOW harbors the respective genes downstream of an IPTG-inducible P*_spac_* promoter, and pGL485 carries a constitutively expressed lacI gene [125]. In order to induce the expression of the genes under the control of P*_spac_*, the bacteria were incubated with 50 µM isopropyl β-D-1-thiogalactopyranoside (IPTG; Sigma, St. Louis, MO, USA) for 2 h according to Bottomley et al. [75]. After induction of protein expression, the bacteria were used for the downstream assays. 

The day before the experiment, the frozen bacterial stock was inoculated in tryptic soy broth (TSB) (Acumedia, Neogen, Lansing, MI, USA) at a ratio of 1:100 and incubated at 37 °C overnight until an OD_600nm_ of 1.8–2.0 was reached. On the following day, the bacteria were resuspended in TSB supplemented with 1% D-glucose (TSBG) to an OD_600nm_ of 0.05–0.3 depending on the assay. For all experiments, the same bacterial culture was used for both control and treated samples that were incubated in parallel for the same time periods with identical initial OD and under the same incubation conditions. Control samples had the same dilutions of ethanol as the treated samples.

### 4.3. Determination of Bacterial Growth

MDRSA CI-M were incubated in 4 mL TSBG in 50 mL tubes in the absence or presence of 50 µg/mL AEA and/or 50 µg/mL methicillin (MET) at an initial OD of 0.1 at 37 °C under constant shaking (150 rpm). At various time intervals, 200 µL of each sample in triplicates were transferred to transparent flat-bottom 96-well plates (Corning Incorporated, Kennebunk, ME, USA), and the OD_600nm_ was measured in a Tecan Infinite M200 microplate reader (Tecan Trading AG, Männedorf, Switzerland). In parallel, 100 µL of each sample in triplicates were transferred to µ-clear flat-bottomed 96-well white plates (Greiner Bio-One GmbH, Frickenhausen, Germany) to which 100 µL of the BacTiter^Glo^ bacterial viability reagent (Promega Corporation, Madison, WI, USA) were added to measure the bacterial ATP content [126]. After a 10 min incubation at room temperature, the luminescence intensity was measured using the Tecan Infinite M200 microplate reader. Moreover, 10 µL of each sample in triplicates was mixed in 990 µL of TSB for further ten-times serial dilutions for determining the colony forming units (CFUs) after spreading 100 µL of each dilution on tryptic soy agar plates (Acumedia, Neogen, Lansing, Michigan, MI, USA) [126]. The CFU of each sample was calculated by the following formula: number of colonies x dilution factor x original volume of sample.

### 4.4. Bocillin-FL Accumulation Assay Using Flow Cytometry and Spinning Disk Confocal Microscopy

MDRSA CI-M at an initial OD_600nm_ of 0.2, was incubated in TSBG in the absence or presence of 50 µg/mL AEA and/or 10 µg/mL Bocillin-FL (Thermo Scientific, Rockford, IL, USA) for 2 h at 37 °C in a shaking incubator (150 rpm). At the end of incubation, the bacteria were washed with PBS and the green fluorescence intensity was measured by flow cytometry (Fortessa LSR Flow cytometer, BD Biosciences, San Jose, CA, USA) with an excitation/emission of 488 nm/530 nm. The BD FACSDiva software was used for the collection of data (20,000 events per sample) and the FCS Express 7 software was used for analyzing the data. 

For visualizing the Bocillin-FL staining of the bacteria, the samples were fixed with 1% paraformaldehyde for 30 min resuspended in DDW, and inspected under a spinning disk confocal microscope (Nikon Yokogawa W1 Spinning Disk with 50 µm pinholes) after spotting the sample on an agarose pad. The Plan-Apochromat ×100 objective was used with the 488 nm excitation laser and the green filter for GFP, and the images were captured using the NIS element software.

### 4.5. Analysis of Bocillin-FL-Bound Proteins on SDS-PAGE

MDRSA CI-M or MSSA ATCC25923 cultures at an initial OD_600nm_ of 0.25 were incubated in a shaker incubator at 37 °C for 2.5 h to reach an OD_600nm_ of 1.0. Thereafter, the cultures were incubated with or without 50 µg/mL AEA for 1 h at 37 °C in a shaker incubator (150 rpm), followed by a 1 h incubation with 10 µg/mL of the fluorescent penicillin Bocillin-FL [127]. At the end of incubation, the culture was centrifuged 21,000× *g* for 5 min at 4 °C, and the bacteria pellet was extracted in B-PER complete bacterial extraction buffer (Pierce, Thermo Scientific, Rockford, IL, USA) supplemented with 50 µg/mL lysostaphin (ProSpec-Tany TechnoGene Ltd., Ness-Ziona, Israel), 1 mM MgSO_4_, 1 mM PMSF and 1× protease inhibitor cocktail (Sigma, St. Louis, MO, USA) at 37 °C for 10 min. After hard vortexing, insoluble material was pelleted and the protein concentration of the soluble fraction was determined by BCA Pierce reagent (Thermo Scientific, Rockford, IL, USA), diluted in reduced 4×Laemmli loading buffer, boiled at 95 °C for 10 min, and run in a 4–15% gradient SDS-PAGE gel (Mini-PROTEAN TGX stain-free gel, Bio-Rad Laboratories, Inc., Hercules, CA, USA). The insoluble membrane fraction was sonicated in 50 µL B-PER lysis buffer for 15 min in an ice-cold sonicator water bath (Transsonic 460, Elma Schmidbauer GmbH, Singen, Germany) and 4× non-reducing Laemmli buffer was added and all samples boiled at 95 °C for 10 min before loading on a 4–15% gradient SDS-PAGE gel. The green fluorescence of Bocillin-FL was detected using the ChemiDoc XRS+ imaging system (Bio-Rad, Hercules, CA, USA). Thereafter the gels were silver stained (Pierce silver stain kit, Thermo Scientific, Rockford, IL, USA) according to the manufacturer’s instructions. 

### 4.6. Fluorescent AEA Accumulation Assays

The uptake assay of fluorescent AEA (SKM 4-45-1; Cayman Chemical, Ann Arbor, MI, USA) [55] into MDRSA CI-M was performed by incubating 200 µL bacterial culture at an OD_600nm_ of 0.3 in the absence (control) or presence of 5 µg/mL fluorescent AEA in PBS supplemented with 1% D-glucose, and the fluorescence intensity measured every 10 min for 2 h at 37 °C in the M200 Infinite plate reader with excitation at 488 nm and emission at 535 nm (Tecan M200 infinite plate reader).

For spinning disk confocal microscopy, MDRSA CI-M at an OD_600nm_ of 0.3 was incubated in the absence or presence of 5 µg/mL fluorescent AEA and/or 50 µg/mL AEA for 2 and 4 h in TSBG. At the end of incubation, the samples were fixed in 1% paraformaldehyde for 20 min, and then visualized using the Plan-Apochromat ×100 objective and the 488 nm excitation laser using the green filter for GFP.

### 4.7. Detection of Exopolysaccharides (EPS) Production by Congo Red

MDRSA CI-M at an OD_600nm_ of 0.1 was incubated for 1 h in the absence or presence of 12.5 or 50 µg/mL AEA in TSBG, and then 3 drops of 10 µL of each sample were applied on a BHI agar plate that contains 0.08% Congo Red with or without 5% sorbitol. The amount of EPS was visualized on the following day by the black staining caused by EPS interaction with Congo Red [128]. 

### 4.8. Detection of Extracellular DNA (eDNA) by Methyl Green Agar Plates

MDRSA CI-M at an OD_600nm_ of 0.3 was incubated in the absence or presence of 25 or 50 μg/mL AEA for 1 h, and then 10 μL of the cultures were inoculated on TSA with or without 0.02% Methyl Green (Sigma, St. Louis, MO, USA) followed by a 24 h incubation at 37 °C. Methyl Green binds to DNA and stains it green-blue [129].

### 4.9. Bacterial DNA Staining with TOTO-1, SYTO 9, and PI

MDRSA CI-M at an OD_600nm_ of 0.3 was incubated in the absence or presence of 25 or 50 μg/mL AEA for 2 h, washed in PBS, and then stained with either 2 μM TOTO-1 iodide (Invitrogen, Life Technologies, ThermoFisher Scientific, Carlsbad, CA, USA) alone or with 2 μM SYTO 9 (Invitrogen, Life Technologies, ThermoFisher Scientific, Carlsbad, CA, USA) together with 2 μg/mL propidium iodide (PI; Sigma, St. Louis, MO, USA) in PBS for 20 min at room temperature, and then analyzed by flow cytometry (Fortessa LSR Flow cytometer, BD Biosciences) using the 488 nm excitation laser and collecting the data using the green (530 nm) filter for TOTO-1 and SYTO 9 and red (610/620 nm) filter for PI [59,130,131]. For each sample, 50,000 events were collected. TOTO-1 stains the eDNA, while SYTO 9 freely diffuses through the bacterial cell wall and thus stains both extracellular and intracellular DNA. PI stains eDNA and can only enter the bacteria after membrane disruption, upon which the intracellular nucleic acids are stained.

### 4.10. Surface Binding of WGA, ConA, and Dextran 10,000

MDRSA CI-M at an initial OD_600nm_ of 0.2 was incubated for 2 h in the absence or presence of 50 µg/mL AEA, and then washed in PBS and incubated with 10 µg/mL of one of the three following fluorescent-conjugated reagents obtained from Invitrogen (ThermoFisher Scientific, Eugene, OR, USA): AlexaFluor^647^-wheat germ agglutinin (WGA) that binds to N-acetyl-D-glucosamine and sialic acid, AlexaFluor^647^-Concanavalin A (ConA) that binds to mannose residues, and AlexaFluor^647^-Dextran 10,000, which is an anionic molecule that binds to positively charged molecules such as the amino acid lysine and extracellular polymeric substances. After 20 min incubation, the bacteria were washed in PBS and analyzed on flow cytometry (Fortessa LSR Flow cytometer, BD Biosciences) with an excitation/emission of 640 nm/670 nm.

### 4.11. Determination of the Cell Wall Teichoic Acid (WTA) Content

The cell wall content of teichoic acid was determined by a modification of the protocols described by Meredith et al. [132] and Covas et al. [133]. An overnight culture of MDRSA CI-M was diluted to an OD_600nm_ of 0.2 in TSBG and let grow to an OD_600nm_ of 0.5 in a shaking incubator (150 rpm) at 37 °C. Then the culture was treated with either 50 µg/mL AEA or an equal amount of ethanol (0.5%) for 2 h. At the end of incubation, the OD_600nm_ was measured for each sample, and 4 OD equivalents were taken from each sample. The bacteria were centrifuged at 4000× *g* for 10 min, and washed with 40 mL 50 mM 2-(N-morpholino)ethanesulfonic acid (MES), pH 6.5, recentrifuged, and resuspended in 1 mL of 4% SDS in 50 mM MES pH 6.5. The samples were autoclaved using the 20 min program at 121 °C to ensure full inactivation of degradative enzymes. After autoclaving, the samples were centrifuged at 21,000× *g* for 10 min at 4 °C. The supernatant served as a loading control that was run on a 10% SDS-PAGE Mini-Protean gel (Bio-Rad Laboratories, Hercules, CA, USA) followed by silver staining (Pierce silver stain kit, Thermo Scientific, Rockford, IL, USA). The pellet was further processed for WTA isolation. The pellet was washed twice with 1 mL 4% SDS in 50 mM MES pH 6.5, then once in 1 mL 2% NaCl in 50 mM MES pH 6.5, once in 1 mL of 50 mM MES pH 6, followed by a 4 h incubation at 50 °C with 1 mL of 20 µg/mL Proteinase K in 20 mM Tris-HCl pH 8.0 containing 0.5% SDS in an MS-100 Thermo-shaker at 1200 rpm (Hangzhou Allsheng Instruments Co., Ltd., Xihu District, Hangzhou, China). At the end of incubation, the samples were centrifuged and the pellets were washed once with 2% NaCl in 50 mM MES pH 6.5 and three times with sterile DDW to remove SDS. The resulting pellets, which consist of the cell wall, were then resuspended in 200 µL of 0.1 M NaOH at 37 °C per 20 mg cell wall pellet and incubated in an MS-100 Thermo-shaker at 1400 rpm for 16 h. At the end of incubation, the supernatant containing the WTA was collected and neutralized with ¼ volume of 1 M Tris pH 7.0, and diluted 1:3 in 3× loading buffer (0.3 M Tris HCl pH 8.2, 0.3 M tricine, 30% glycerol, 0.15% bromophenol blue). Thirty µL of the samples was run on a 20% native 16 cm-long PAGE gel in 1 M Tris HCl buffer pH 8.5 using the Hoefer gel system (Hoefer Scientific Instruments, San Francisco, CA, USA) and 0.1 M Tris HCl and 0.1 M Tricine pH 8.2 as running buffer. The gel was run for 24 h at 40 mA at 4 °C. The gel was washed in DDW, and then stained with 0.1% Alcian blue in 5% acetic acid and 40% ethanol, followed by silver staining (Pierce, Thermo Scientific, Rockford, IL, USA). The stained gels were imaged by the ChemiDoc XRS+ imaging system (Bio-Rad, Hercules, CA, USA) and also captured by the camera of a Samsung smartphone.

### 4.12. Laurdan Membrane Fluidity Assay

Laurdan staining was performed by a slight modification of the protocols described by Wenzel et al. [66] and Stahl et al. [134]. MDRSA CI-M at an initial OD_600nm_ of 0.3 was incubated in the absence or presence of increasing concentrations of AEA for 2 h at 37 °C in a shaking incubator (150 rpm). Then, Laurdan (AnaSpec Inc., Fremont, CA, USA) was added to a final concentration of 10 µM, and the samples were incubated for 10 min at 37 °C. Parallel unstained samples were prepared. At the end of incubation, the bacteria were washed four times with 1 mL PBS that was supplemented with 1% D-glucose and 0.1% dimethylsulfoxide (DMSO) (PBSGD). The samples were then adjusted to the same OD with PBSGD. The fluorescence emitted from 200 µL of each sample in triplicates in a µ-clear black flat 96-well plate (Greiner Bio-One) was monitored in the M200 infinite plate reader (Tecan, Trading AG, Männedorf, Switzerland) with an excitation of 350 nm and emission spectrum from 400 nm to 600 nm at 30 °C. The overall membrane fluidity was measured using Laurdan generalized polarization (GP) values according to the formula GP= (RFI_440nm_ − RFI_490nm_)/(RFI_440nm_ + RFI_490nm_) as described [66].

### 4.13. 10-N-Nonyl-Acridine Orange (NAO) Staining of Cardiolipin 

Control and AEA (2 h)-treated MDRSA CI-M were exposed to 1 µM NAO (Santa Cruz Biotechnology, Dallas, TX, USA) for 30 min at 37 °C, and the green and red fluorescence intensities were analyzed by flow cytometry (Fortessa LSR Flow cytometer, BD Biosciences) using the 488 nm excitation laser and the green (530 nm) and red (610/620 nm) filters. Upon binding of NAO to cardiolipin, the fluorescence emission shifts from green to red [72].

### 4.14. Two-Dimensional Thin-Layer Chromatography of Membrane Lipids

The pellets (5 OD units) of control and AEA (50 μg/mL, 2 h)-treated MDRSA CI-M were resuspended in 100 μL of DDW and sonicated on ice with a probe sonicator (UP200S Ultrasonic homogenizer, Hielscher Ultrasonics, Teltow, Germany) for 1 min with a 30% amplitude and 0.5 s intervals. The lipids from the sonicated samples were extracted according to the Bligh and Dyer method [135], and loaded onto a TLC plate (TLC Silica gel 60 F254 20 × 20 cm, Merck, Darmstadt, Germany). As controls, 5 μg of cardiolipin (CL) (Avanti Polar Lipids, Alabaster, AL, USA) and 5 μg of phosphatidylethanolamine (PE) (Sigma, St. Louis, MO, USA) were loaded on the TLC plates. For the first dimension, the plates were run in chloroform-methanol-water solution (65:25:4, *v*/*v*/*v*), and for the second dimension, they were run in chloroform-methanol-acetic acid-water solution (90:15:10:3.5, *v*/*v*/*v*/*v*). Then, the plates were dried for 5 min and sprayed with Molybdenum Blue reagent (Sigma, St. Louis, MO, USA). Five minutes after spraying, the images of the developed TLC plates were captured using the Fusion Solo S Imager (Vilber, Marne-la-Vallée Cedex 3, France) and the Fusion.CAPT software. 

### 4.15. Membrane ATPase and GTPase Assay

Quantification of membrane ATPase activity was performed by a slight modification of the protocol described by Hoelscher and Hudson [136]. An overnight culture of MDRSA CI-M was diluted in TSBG to an OD_600nm_ of 0.2 and incubated with constant shaking (150 rpm) at 37 °C until reaching an OD_600nm_ of 1. The culture was then incubated with 12.5 or 50 µg/mL AEA or respective ethanol concentrations that served as controls for 30 min at 37 °C under constant shaking (150 rpm). Five OD_600nm_ units were taken from each sample and centrifuged at 4000× *g* for 10 min at 4 °C. The cells were washed once with 10 mL of 50 mM Tris-maleate buffer pH 6.0, and then resuspended in 1 mL of 50 mM Tris-maleate buffer pH 6.0 to which 100 µL of toluene was added to perforate the bacteria. After a vigorous vortex, the samples were freeze-thawed three times in liquid nitrogen/37 °C, each step for 5 min. The permeabilized cells were harvested by centrifugation at 5000× *g* for 5 min at 4 °C, then washed four times with 1 mL of 50 mM Tris-maleate buffer pH 6.0 and twice with 1 mL sterile DDW. Then, the membrane pellets were resuspended in ice-cold 1 mL of 50 mM Tris-Maleate buffer pH 6.0 supplemented with 10 mM MgSO_4_ to which ATP was added to a final concentration of 7.5 mM. Thereafter, the samples were incubated at 37 °C. One hundred and twenty µL of each sample was harvested at various time points including time 0 to determine the phosphate content which reflects the ATPase activity. The harvested samples were centrifuged at 21,100× *g* at 4 °C to get clear supernatants. Twice 50 µL of each supernatant was transferred to the wells of ice-cold transparent flat-bottomed 96-well plates that were kept at −20 °C until all samples were collected. The phosphate levels were measured by adding 100 µL of the BIOMOL Green reagent (Enzo Life Sciences, Farmingdale, NY, USA) and reading the absorbance at 620 nm after a 15 min incubation. The phosphate concentrations were calculated against a standard curve made by known concentrations of phosphate. The same protocol was used to measure GTPase activity, but instead of ATP, 1.5 mM GTP (Sigma, St. Louis, MO, USA) was used.

### 4.16. Autolysis Assay

The autolysis experiment was performed with a slight modification of the protocol of Tiwari et al. [84]. An overnight culture of MDRSA CI-M was diluted to an OD_600nm_ of 0.3 in TSBG, and then exposed to 50 µg/mL AEA or equal ethanol concentration (0.5%) as a control for 2 h at 37 °C. At the end of incubation, the bacteria were washed 4 times in 50 mM Tris buffer pH 7.6 and resuspended to an OD_600nm_ of 1. The bacteria were then incubated in 50 mM Tris buffer pH 7.6 in the absence or presence of 0.05% Triton X-100, and the optical density at 600 nm was measured every 10 min at 37 °C for 6 h in an M200 Infinite plate reader (Tecan, Trading AG, Männedorf, Switzerland).

### 4.17. Autolysin Zymogram

Autolysin zymogram was performed according to Vaz and Filipe [89]. Cell wall substrate was prepared from MDRSA CI-M. To this end, MDRSA CI-M at an initial OD_600nm_ of 0.05 in 200 mL TSB was grown at 37 °C in a shaker incubator (200 rpm) until reaching an OD_600nm_ of 1. Then, the bacteria were centrifuged at 4000× *g* for 10 min at 4 °C and the bacterial pellet washed with 200 mL sterile DDW. After recentrifugation, the bacteria were resuspended in 24 mL sterile DDW and autoclaved for 20 min at 121° C. After cooling down, the cell wall was precipitated at 21,100× *g* for 10 min at 4 °C, and DDW was added to obtain a 50 mg/mL cell wall suspension that was used to prepare the 10% zymogram SDS-PAGE gel containing a final concentration of 1 mg/mL *S. aureus* cell wall.

For the preparation of the crude autolytic extracts, an overnight culture of MDRSA CI-M was diluted to an OD_600nm_ of 0.02 in TSBG and let grew to an OD_600nm_ of 0.3 in a shaking incubator (200 rpm) at 37 °C. Then, the bacteria were incubated in the absence or presence of 50 µg/mL AEA for 2 h in a shaking incubator (200 rpm) at 37 °C in 5 mL TSBG. At the end of incubation, the OD_600nm_ was measured and equal OD equivalents were taken from the samples and placed in ice-cold water. The bacteria were centrifuged at 4000× *g* for 10 min at 4 °C, and washed twice with 40 mL ice-cold washing buffer composed of 50 mM Tris HCl pH 7.6 and 150 mM NaCl. Then the pellet was resuspended in 50 µL of 4% SDS in sterile DDW and incubated at 25 °C for 30 min. The supernatant that contains the autolysins was collected and the protein content was determined in a nano-drop (OD at λ = 280 nm). The samples were diluted in non-reduced 4× Laemelli buffer and loaded in parallel to a zymogram and a regular 10% SDS-PAGE. At the end of the run, the zymogram gel was washed four times for 15 min in DDW to remove SDS, and then incubated in renaturation buffer (50 mM Tris HCl, pH 7.6, 0.1% Triton X-100, 10 mM CaCl_2_, 10 mM MgCl_2_) at 37 °C in a shaking incubator (50 rpm) for 24 h. Then, the zymogram was stained in 0.1% methylene blue solution in 0.01% potassium hydroxide for 1 h at room temperature under gentle shaking, followed by several washes in DDW. Empty bands appeared in the gel where the autolysins were.

### 4.18. Gelatin and Casein Zymogram

The preparation of gelatin and casein zymograms was based on the protocol of Duanis-Assaf et al. [137] with adapted modifications. Control, AEA (50 μg/mL, 2 h) and/or MET (50 μg/mL, 2 h)-treated MDRSA CI-M were washed twice in 1 mL 50 mM Tris HCl pH 7.6 to remove the growth media, and then resuspended in 100 μL 4× non-reduced Laemelli buffer (Bio-Rad Laboratories, Hercules, CA, USA) per 10 OD units. The samples were subjected to 15 min sonication in an ice-cold sonicator water bath (Transsonic 460, Elma Schmidbauer GmbH, Singen, Germany) followed by another 30 min at room temperature with intermittent vortexing. Thirty μL was loaded on a 16 cm-long non-reduced SDS-PAGE composed of 5%, 7.5%, and 15% gel at a ratio of 1:1:1 (Hoefer apparatus), or 10 μL was loaded on a 7 cm-long non-reduced SDS-PAGE composed of 7.5% and 15% gel at a ratio of 1:1 using the Bio-Rad Mini-Protean system. Gelatin (Sigma) and casein (Sigma) were incorporated into the gels at a final concentration of 1.2 mg/mL. This system made it possible to simultaneously catch high- and low-molecular-weight proteases. The samples were not heated before loading and the gels were run at 4 °C. At the end of electrophoresis, the gels were washed twice in ddw for 15 min at room temperature, followed by two washes of 30 min with 2.5% Triton X-100 in 50 mM Tris pH 7.6, 5 mM CaCl_2_, 1 mM ZnCl_2_, and 1 mM MgSO_4_ at room temperature to remove SDS and renaturate the proteins, and 18 h incubation in 1% Triton X-100 in 50 mM Tris pH 7.6, 5 mM CaCl_2_, 1 mM ZnCl_2_, and 1 mM MgSO_4_ at 37 °C under constant shaking (50 rpm). Thereafter, the gels were incubated in 50 mM Tris pH 7.6, 5 mM CaCl_2_, 1 mM ZnCl_2_, and 1 mM MgSO_4_ at 37 °C for another 24 h. At the end of incubation, the gels were stained with 0.25% Coomassie blue in 10% acetic acid, 40% methanol, and 50% DDW for 1 h, and destained in 10% acetic acid, 40% methanol, and 50% DDW. Images were captured by the camera of a Samsung smartphone and by using the Fusion Solo S imaging system. 

### 4.19. Protease Assay on Gelatin Agar Plates

Ten μL of control and AEA (50 μg/mL, 2 h)-treated MDRSA CI-M cultures at an OD_600nm_ of 0.3 were inoculated on tryptic soy agar plates (Acumedia, Neogen, Lansing, MI, USA) containing 1.5% gelatin followed by a 24 h incubation at 37 °C. Alternatively, 10 μL of their supernatants were inoculated on these plates. The clearance zone around the inoculation site was detected after staining the plates with 0.1% crystal violet solution prepared from a 0.4% Gram’s crystal violet solution (Merck, EMD Millipore Corporation, Billerica, MA, USA) diluted with DDW. After a 15 min incubation, the plates were washed with DDW, and images were captured using the Bio-Rad ChemiDoc XRS+ Imager system and the Image Lab software. 

### 4.20. RNA and cDNA preparation 

The RNA preparation was performed by a slight modification of the protocol described by Banerjee et al. [49]. MDRSA CI-M at an initial OD_600nm_ of 0.1 was incubated under constant shaking (150 rpm) at 37 °C for 2 h to reach an OD of 0.5. The culture was then incubated in TSBG with 50 µg/mL AEA or respective ethanol concentration (0.5%) that served as control for 2 h at 37 °C under constant shaking (150 rpm). The cultures were harvested by centrifugation at 4000× *g* for 10 min at 4 °C, washed once with 10 mL of 50 mM MES buffer pH 6.5, and then incubated for 10 min at 37 °C with 100 µL of UPW containing 100 µg/mL lysostaphin (ProSpec-Tany TechnoGene Ltd., Ness-Ziona, Israel). Immediately at the end of incubation, 1 mL of Tri-Reagent (Sigma-Aldrich) was added to each tube, vigorously vortexed, and transferred to NucleoSpin beads tubes type B (Macherey-Nagel, GmbH&Co, Düren, Germany). The bacteria were then broken by inserting the tubes into the FP120 FastPrep cell disruption system (BIO 101, Savant Instruments, Holbrook, NY, USA) operating three times for 45 s at a speed of 4.5, with a 5 min break on ice between each cycle. After removing the beads, 200 µL of chloroform was added to the Tri-Reagent supernatant, followed by vigorous vortexing for 15 sec. The samples were allowed to stand 15 min at room temperature before centrifugation at 21,100× *g* for 15 min at 4 °C. The upper water phase was collected to which an equal volume of isopropanol was added to precipitate the RNA. After 30 min incubation at room temperature, the RNA was precipitated by a 30 min centrifugation at 21,100× *g* for 30 min, followed by two washes of the RNA pellet with 1 mL 75% EtOH, and then the pellet was let dry for 30 min before resuspending in DNase- and RNase-free water (Bio-Lab Ltd., Jerusalem, Israel).

The quality of the RNA was determined by running the samples in 1% agarose with ethidium bromide, and the quantity was determined by a Nanodrop instrument. DNase treatment was performed when required using the PerfeCTa RNase-free DNaseI (Quanta Biosciences, Beverly, MA, USA) according to the manufacturer’s instructions. RNA was transcribed to cDNA using the AB high-capacity cDNA reverse transcription kit (Applied Biosciences by ThermoFisher Scientific, Vilnius, Lithuania). The RNA (2 µg) was first mixed with the random primers and incubated for 5 min at 70 °C followed by 2 min incubation at 4 °C. Thereafter, the reaction buffer, RNase inhibitor, and reverse transcriptase were added, and the reaction was performed at 42 °C for 2 h followed by inactivation of the enzyme for 5 min at 85 °C.

### 4.21. Quantitative Real-Time PCR

Real-time qPCR was performed in a CFX96 Bio-Rad Connect Real-Time PCR apparatus using Power Sybr Green Master Mix (Applied Biosystems, Life Technologies) on 10 ng cDNA in the presence of 300 nM forward/reverse primer sets (Appendix A). PCR conditions included an initial heating at 50 °C for 2 min, an activation step at 95 °C for 10 min, followed by 40 cycles of amplification (95 °C for 15 s, 60 °C for 1 min). Calculations of fold-change in gene expression in the treated bacteria in comparison with control bacteria were performed according to the 2^−ΔΔCt^ method [49], where *gyrA*, *gyrB*, *glyA*, *gmk*, *proC*, *recF*, *rho*, *rpoB*, and *asnC* were used as internal standards. These housekeeping genes were tested against each other and found to provide a fold value of 1. In each experiment, two treated samples were tested against each of the two controls using all nine internal gene standards, and from these data, average and standard deviation were calculated. The functions of the different genes studied are described in Appendix A.

### 4.22. Statistical Analysis

The experiments were performed in experimental triplicates and repeated 2–4 times. The data are expressed as the average ± standard error. Statistical analysis was performed using the Microsoft Excel software. The student’s *t*-test was used to compare control and treated samples, with a *p*-value less than 0.05 considered significant.

## 5. Conclusions

The present study complements our previous report on AEA-mediated sensitization of MDRSA to antibiotics [49]. AEA increases the intracellular concentration of not only norfloxacin and the efflux pump substrates ethidium bromide and DAPI, but also of penicillin, which makes the drug efflux inhibition more universal. This is also in light of the increased susceptibility to diverse classes of antibiotics including gentamicin and tetracycline [50]. One of the mechanisms seems to be altered membrane structure that results in reduced membrane ATPase activity thereby inhibiting not only antibiotic efflux, but also membrane transport of WTA, autolysins, and exopolysaccharides. Together with membrane depolarization, these effects lead to inhibition of cell growth, inhibition of biofilm formation and sensitization of the MDRSA to antibiotics (Figure 11).

## Figures and Tables

**Figure 1 ijms-23-07798-f001:**
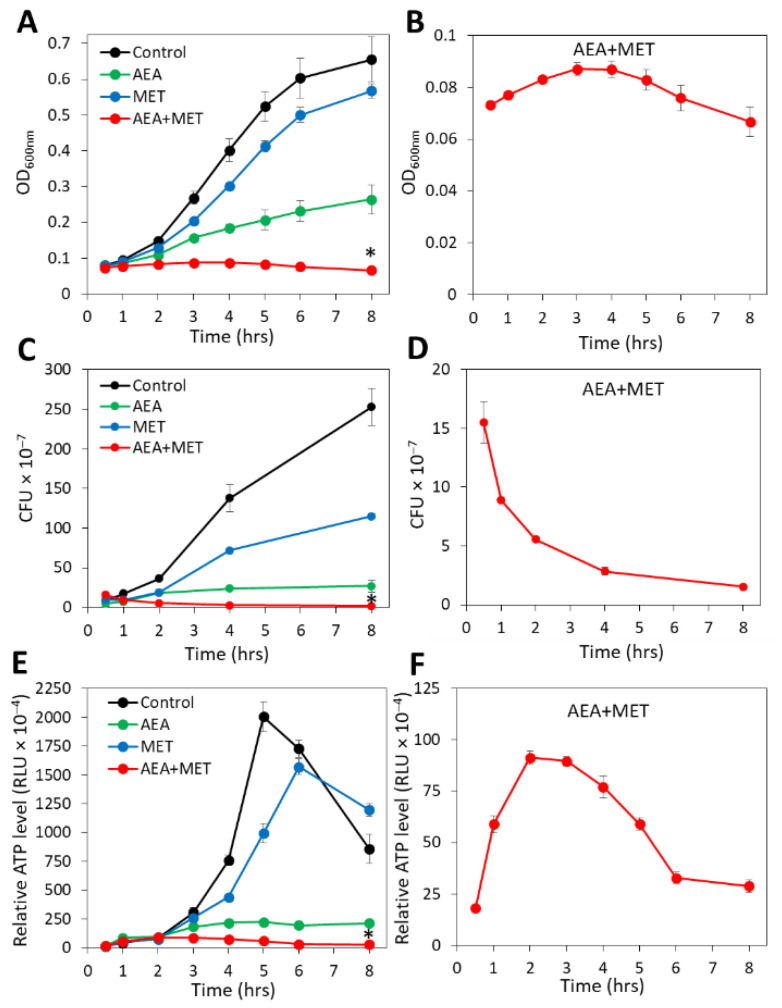
**The combined treatment of AEA with methicillin (MET) caused a reduction in the CFUs with a concomitant transient increase in the ATP content.** MDRSA CI-M were exposed to 50 µg/mL AEA and/or 50 µg/mL MET or equal ethanol concentrations that served as controls. The samples performed in triplicates were analyzed in parallel for changes in turbidity (OD at 600 nm). (**A**,**B**), colony forming units (CFUs) (**C**,**D**) and ATP content (**E**,**F**). (**A**,**C**,**E**) show the four treatment groups together, while the scale of the Y-axis in B, D, and F was adjusted to visualize the changes in the combined AEA/MET treatment group. * *p* < 0.05 when comparing the AEA + MET with either compound alone. RLU = Relative Luminescence Units.

**Figure 2 ijms-23-07798-f002:**
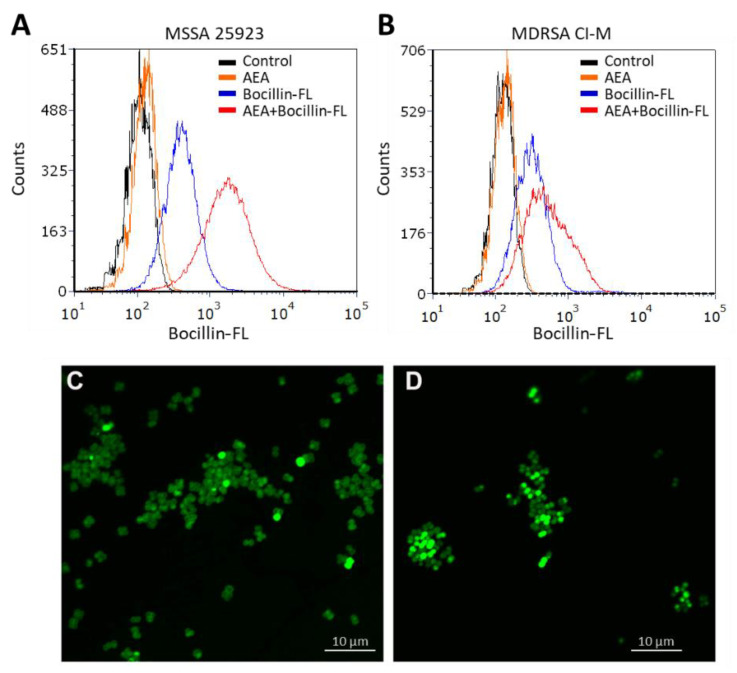
**AEA increases the intracellular level of the penicillin analog Bocillin-FL both in methicillin-sensitive (ATCC 25923) and methicillin-resistant *S. aureus* (MDRSA CI-M).** (**A**,**B**). Flow cytometry of *S. aureus* ATCC 25923 (**A**) and MDRSA CI-M (**B**) that have been incubated with 10 µg/mL Bocillin-FL in the absence or presence of 50 µg/mL AEA for 1 h in TSBG. The green fluorescence of Bocillin-FL was read using the excitation 488 nm laser and emission 520 nm filter. (**C**,**D**). Spinning disk confocal microscopy of MDRSA CI-M that has been incubated with 10 µg/mL Bocillin-FL in the absence (**C**) or presence (**D**) of 50 µg/mL AEA for 2 h. The green fluorescence was visualized using the 488 nm excitation laser and the green filter.

**Figure 3 ijms-23-07798-f003:**
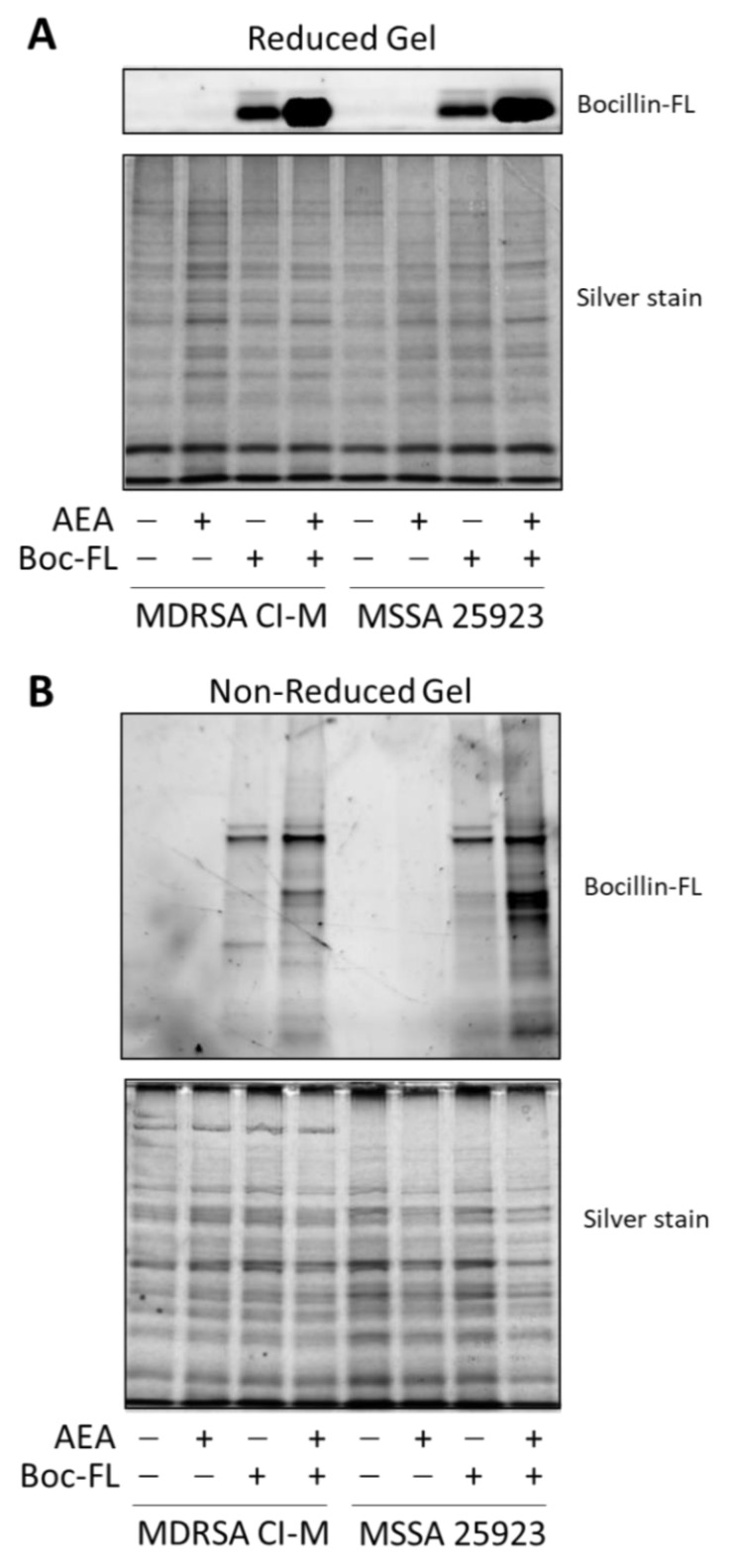
**AEA increases the intracellular concentration of Bocillin-FL with enhanced binding of Bocillin-FL to membrane proteins.** Protein extracts were prepared from MDRSA CI-M and MSSA 25923 that were incubated in the absence or presence of 50 µg/mL AEA for 1 h followed by a 1 h incubation with 10 µg/mL Bocillin-FL. (**A**). The soluble fraction was run on a 10% SDS-PAGE gel under reducing conditions. Upper panel shows the green fluorescence of Bocillin-FL and the lower panel the gel after silver staining. The free Bocillin-FL ran at the front of the gel. (**B**). The membrane fraction was run on a 10% SDS-PAGE gel under non-reducing conditions. Upper panel shows the green fluorescence of Bocillin-FL bound to membrane proteins and the lower panel the gel after silver staining.

**Figure 4 ijms-23-07798-f004:**
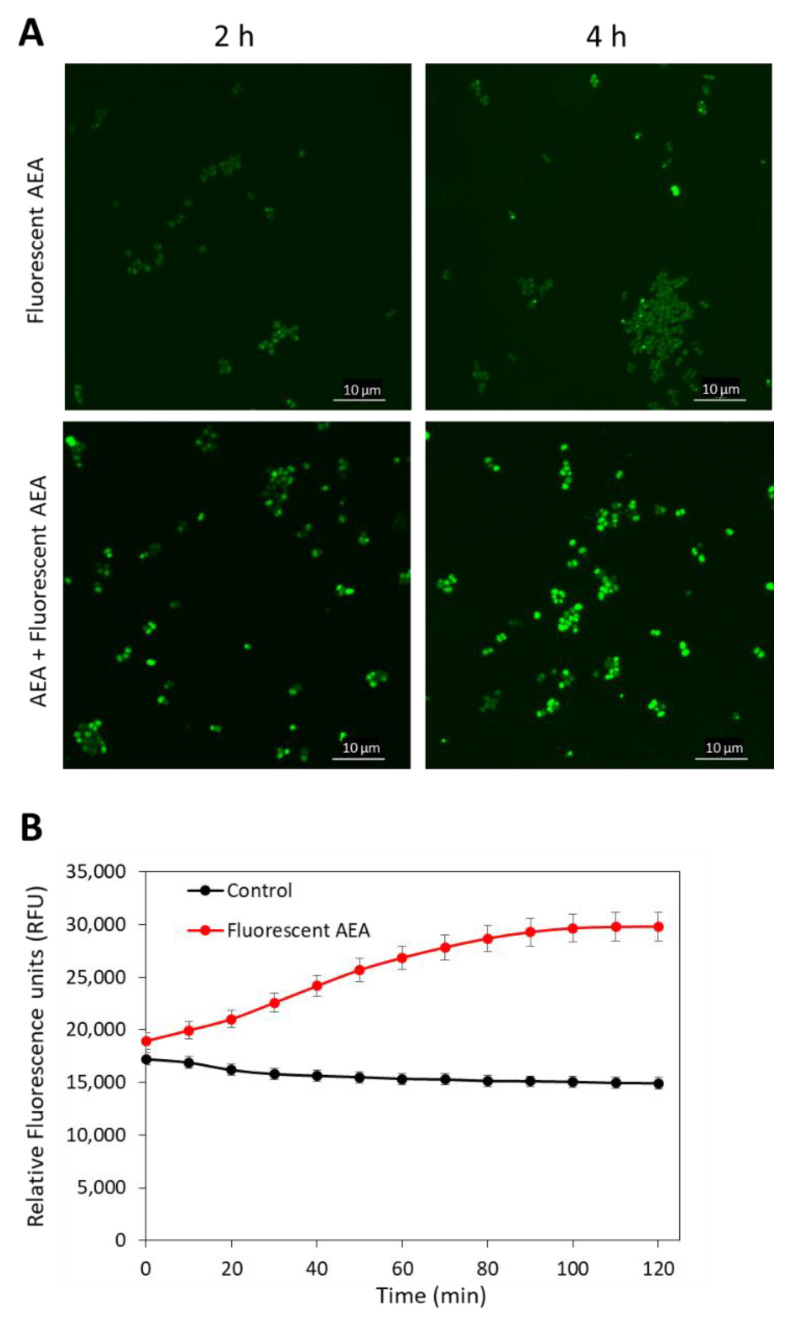
**AEA increases the intracellular concentration of fluorescent AEA.** (**A**). Spinning disk confocal microscopy of MDRSA CI-M that has been treated with 5 µg/mL fluorescent AEA alone (two upper panels) or in the presence of 50 µg/mL AEA (two lower panels) for 2 and 4 h. The green fluorescence was visualized using the 488 nm excitation laser and the green filter. (**B**). MDRSA CI-M was either untreated or exposed to 5 µg/mL fluorescent AEA and incubated in PBS supplemented with 1% D-glucose. The fluorescence intensity was measured with excitation/emission of 488 nm/530 nm each 10 min for 2 h in a M200 infinite Tecan plate reader.

**Figure 5 ijms-23-07798-f005:**
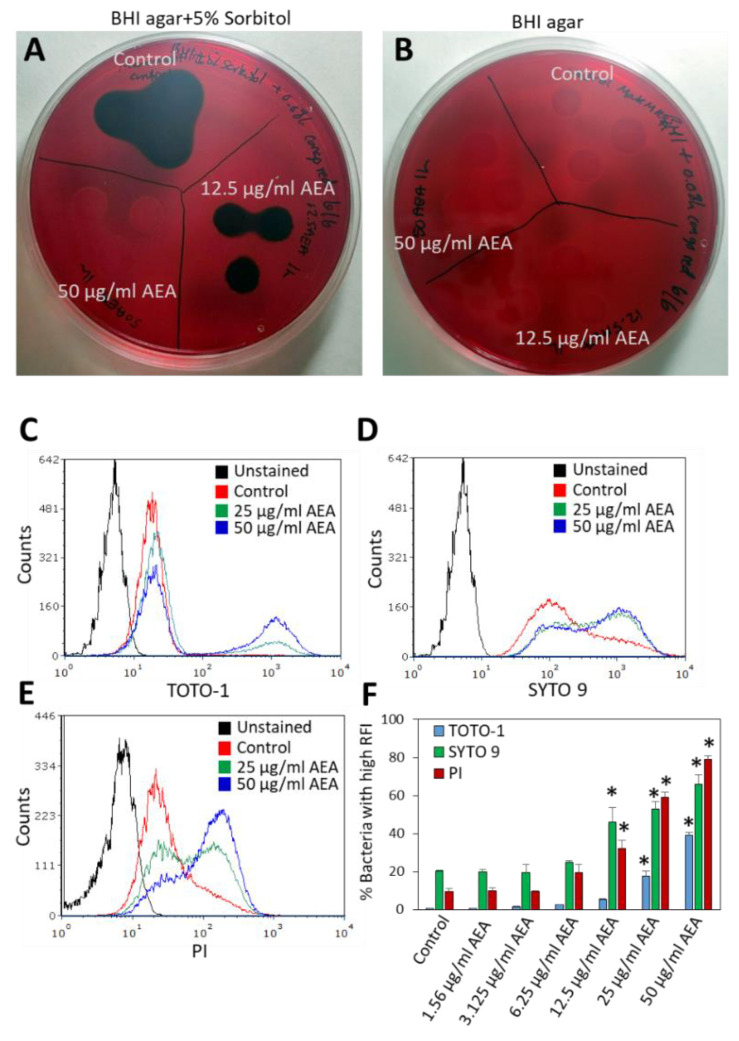
**AEA prevents the secretion of exopolysaccharides (EPS) and increases extracellular DNA (eDNA) production.** (**A**,**B**). MDRSA CI-M was incubated in the absence or presence of 12.5 µg/mL or 50 µg/mL AEA for 1 h, and then 10 µL of each triplicate were placed on Congo Red agar plates containing 5% sorbitol (**A**) or plain Congo red plates (**B**). (**C**,**E**). MDRSA CI-M was incubated in the absence or presence of 12.5 µg/mL or 50 µg/mL AEA for 2 h, and then stained with 2 μM TOTO-1 (**C**), 2 μM SYTO 9 (**D**), or 2 μg/mL PI (**E**) for 20 min and analyzed by flow cytometry. (**F**). The percentage of bacteria with high relative fluorescence intensity (RFI) of samples presented in (**C**,**E**). * *p* < 0.05 when comparing the AEA-treated bacteria with control bacteria.

**Figure 6 ijms-23-07798-f006:**
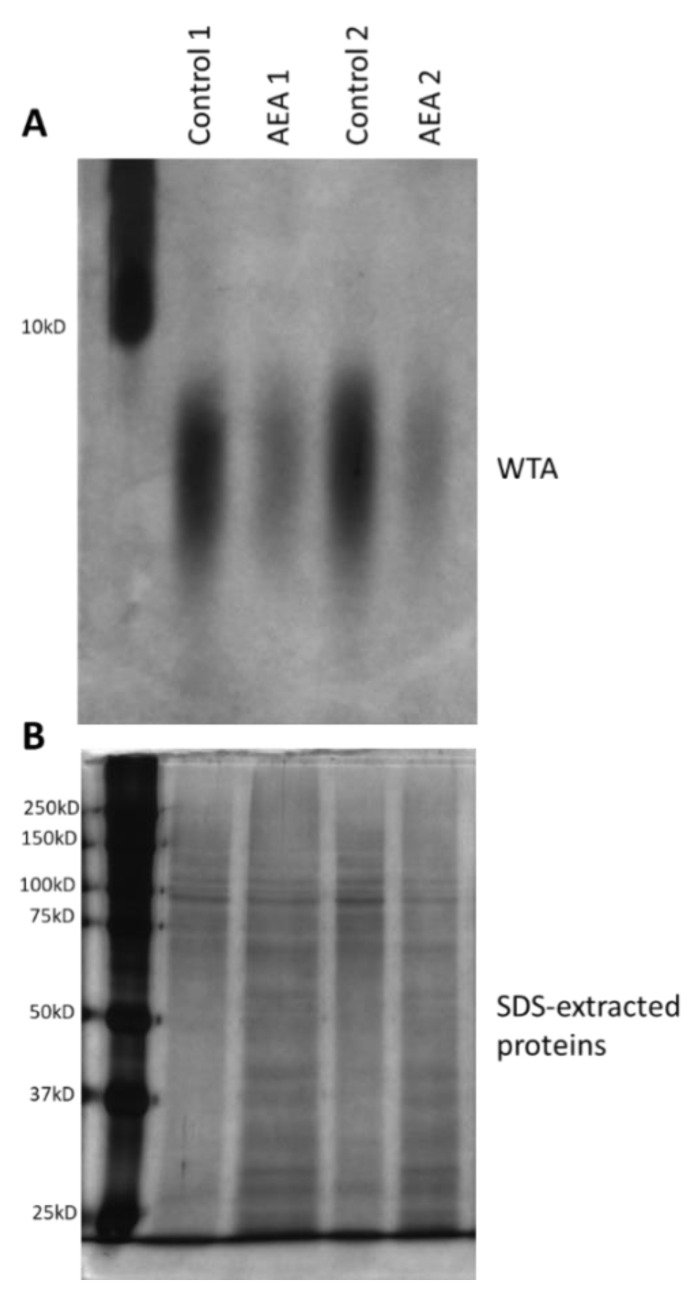
**AEA reduces the cell wall content of wall teichoic acid (WTA).** (**A**). MDRSA CI-M was incubated in the absence or presence of 50 µg/mL AEA for 2 h, and the WTA content analyzed on a 20% PAGE followed by Alcian blue–silver staining. Duplicate samples are presented. (**B**). Silver staining of SDS-extracted proteins run on a 10% SDS-PAGE from the same samples as in (**A**), but from a processing step prior to WTA isolation.

**Figure 7 ijms-23-07798-f007:**
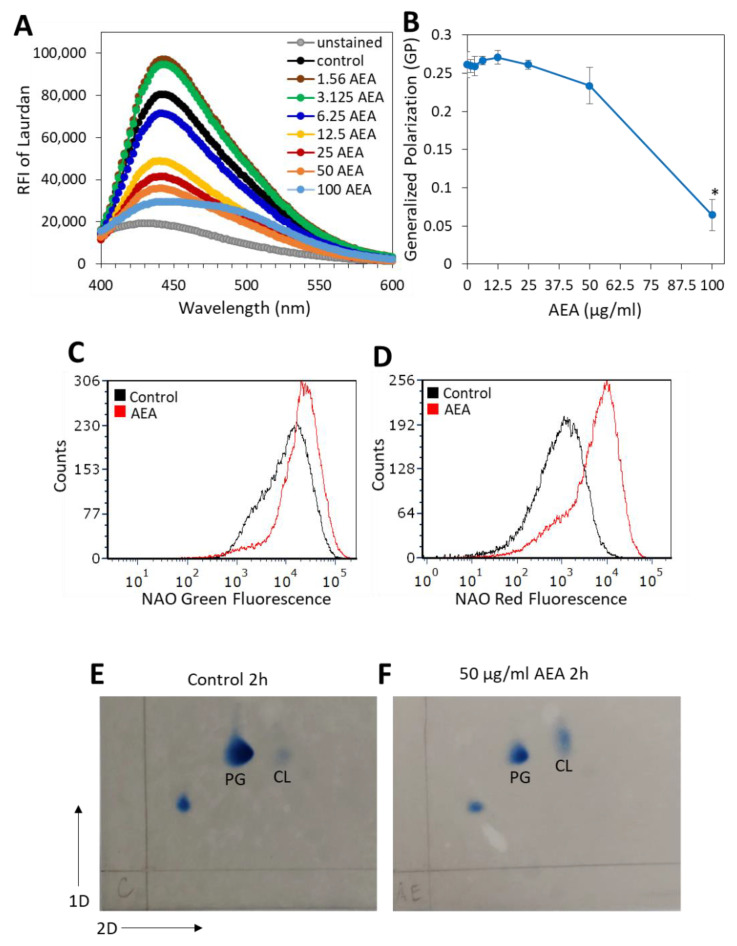
**AEA treatment prevents the incorporation of Laurdan into the bacterial membrane and increases the relative cardiolipin content.** (**A**). MDRSA CI-M was treated with various concentrations of AEA for 5 h and then loaded with 10 µM Laurdan for 10 min at 37 °C, followed by extensive washes in PBS, and the relative emission fluorescence intensities (RFIs) were measured from 400–600 nm upon excitation at 350 nm. (**B**). Generalized Polarization (GR) values of samples in (**A**), calculated according to the formula: GP = (RFI_440nm_ − RFI_490nm_)/(RFI_440nm_ + RFI_490nm_). * *p* < 0.05 when comparing the AEA-treated bacteria with control bacteria. (**C**,**D**). MDRSA CI-M was treated with 12.5 μg/mL of AEA for 2 h at 37 °C and then stained with 10-N-nonyl-acridine orange (NAO) for 30 min at 37 °C, followed by flow cytometry measuring the green (total dye uptake) (**C**) and red fluorescence (dye bound to cardiolipin) (**D**). (**E**,**F**). Thin-layer chromatography (TLC) of membrane lipids isolated from control (**E**) and AEA (50 μg/mL, 2 h)-treated MDRSA CI-M (**F**). PG = Phosphatidylglycerol; CL = Cardiolipin.

**Figure 8 ijms-23-07798-f008:**
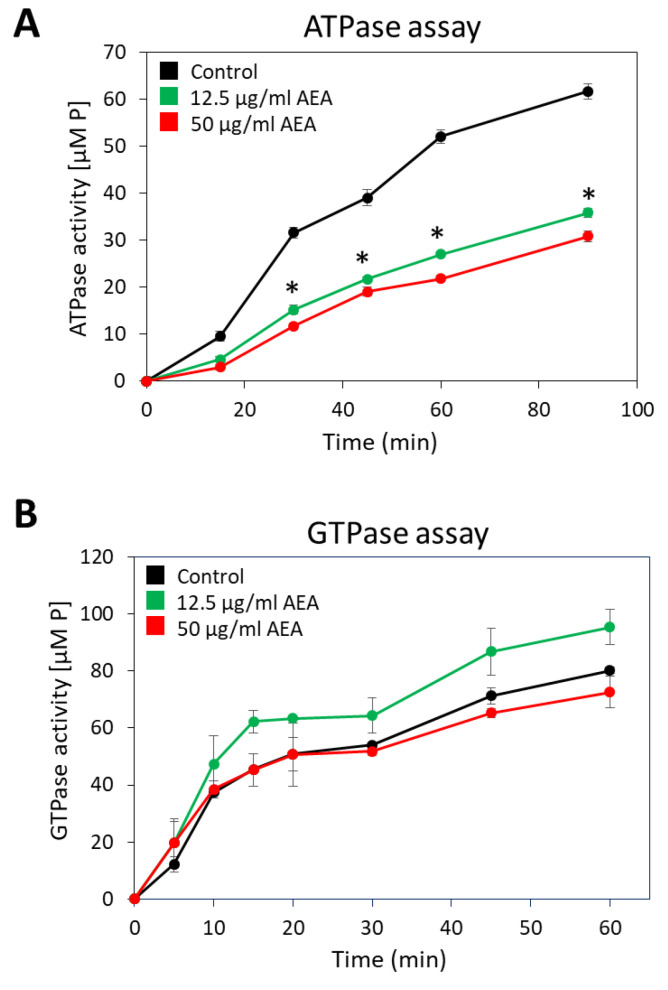
**AEA treatment reduces membrane ATPase activity, but not that of GTPase.** (**A**,**B**). A MDRSA CI-M culture that had reached an OD_600nm_ of 1 was incubated in the absence or presence of 12.5 or 50 μg/mL AEA for 30 min, and the ATPase (**A**) or GTPase (**B**) activities of the isolated membranes were analyzed in a kinetic study using ATP or GTP as substrate, and the concentration of the released phosphate ([P]) was measured using the BIOMOL Green reagent. * *p* < 0.05 when comparing the AEA-treated bacteria with control bacteria.

**Figure 9 ijms-23-07798-f009:**
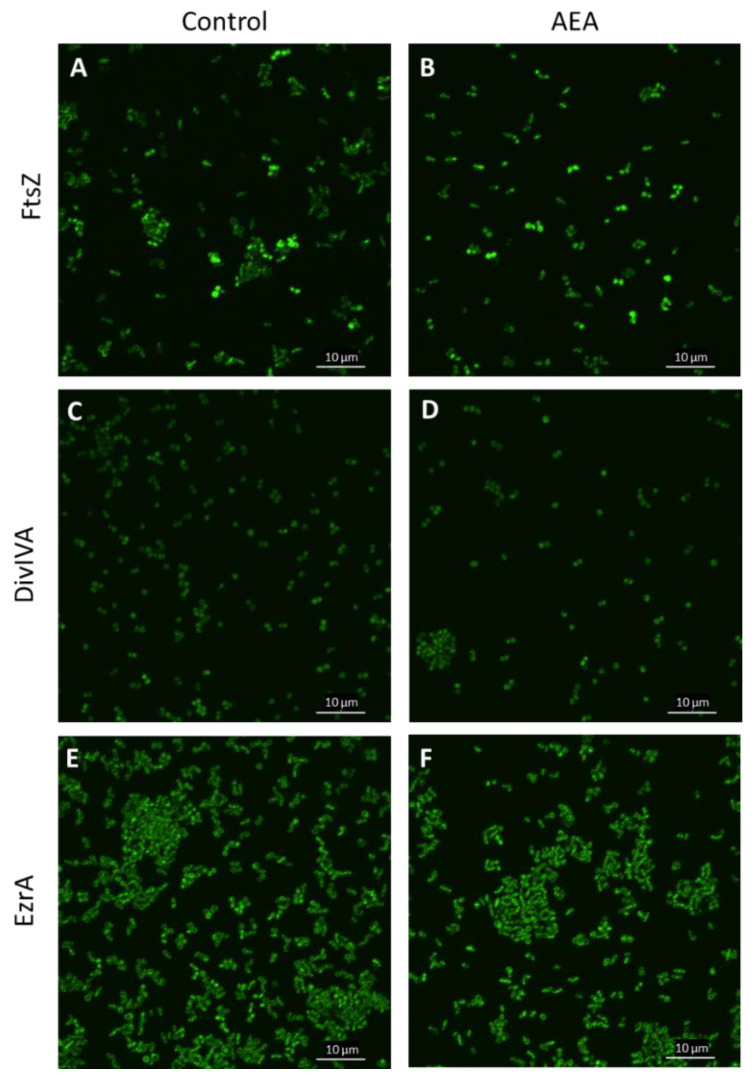
**AEA did not interfere with the formation of the FtsZ Z-ring.** (**A**–**E**). LH607 with inducible FtsZ-GFP (SA103) (**A**,**B**), LH607 expressing inducible DivIA-GFP (SA356) (**C**,**D**), or LH607 expressing EzrA-GFP (SA353) (**E**,**F**) were exposed to 50 μM IPTG for 2 h to induce the expression of the respective GFP fusion proteins and then incubated in the absence or presence of 50 μg/mL AEA for 2 h and the green fluorescence visualized by spinning disk confocal microscopy using the 488 nm excitation laser and the green filter. The bars represent 10 µm.

**Figure 11 ijms-23-07798-f011:**
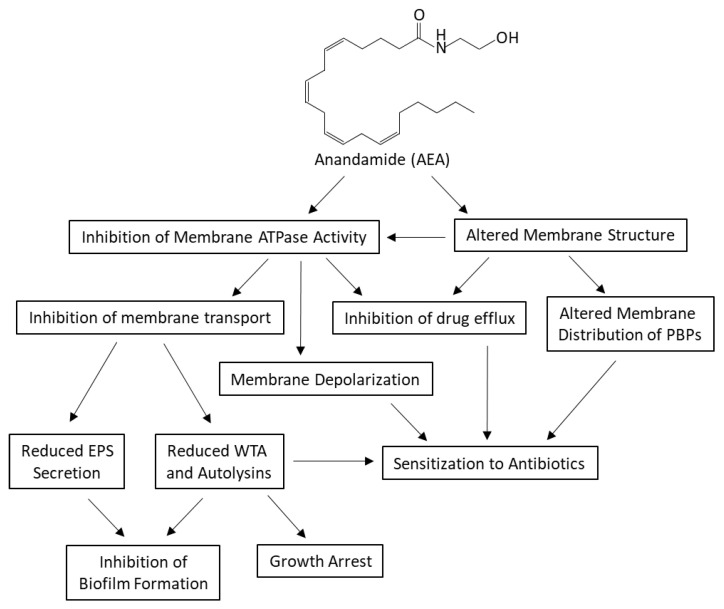
**A summary of the characterized effects of AEA on multidrug resistant *S. aureus*.** AEA is a bioactive lipid conjugate of the unsaturated arachidonic acid and ethanolamine. It causes alterations in the membrane structure and inhibition of membrane ATPase activities. As a result, the membrane distribution of PBPs is altered, the membrane undergoes depolarization, and the drug efflux is inhibited. The reduced membrane ATPase activity also leads to the inhibition of membrane transport of exopolysaccharides (EPS), cell wall teichoic acid (WTA) and autolysins. Altogether, the multiple alterations caused by AEA cumulate in growth arrest, sensitization to antibiotics, and inhibition of biofilm formation.

## Data Availability

Raw data available upon reasonable request.

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
