# Peer review of "Targeting the Achilles’ Heel of Multidrug-Resistant Staphylococcus aureus by the Endocannabinoid Anandamide"

_ijms, 2022, doi:10.3390/ijms23147798_

Round 1
Reviewer 1 Report
The present study has looked at processes that are known to affect biofilm development and antibiotic sensitivity. These include penicillin-binding proteins and their membrane distribution, membrane fluidity and composition, exopolysaccharide (EPS) and extracellular DNA (eDNA) production, cell wall teichoic acid content, membrane ATPase activity, Z-ring divisome formation, and murein hydrolase activities. The paper is very long with too many details and discussion
1- Follow the journal guidelines for abstract writing. It is very long
2- Please try to shorten the materials and methods and provide a rationale for each experiment
3- Did you determine the minimal inhibitory concentration and the minimal bactericidal concentration from combining both drugs together.
4- why do not grow your bacteria for a longer time, more than 8hrs as the drugs might kill the bacteria.
5- Figures 1 B, D, F. Add the controls for the figures.
6- You need to keep the results only in the result section, any comparisons from previous publications can move to the discussion section with citations.
7- In Figures 3 and 6. Is the loaded amount and concentration on the gel similar in treated and control samples?
8- Give more focus on the results obtained by this study, not your previous studies.
Author Response
We thank the Reviewer for reading and critically reviewing the manuscript.
1- Follow the journal guidelines for abstract writing. It is very long
We have now shortened the abstract.
2- Please try to shorten the materials and methods and provide a rationale for each experiment
The rational of each experiment is provided at the beginning of each Result subsection. The study involves several different biochemical and molecular methods that have been adapted from various published papers as cited in the text. These methods were modified to adapt to the specific aims of the present study. Therefore, it was important to provide the exact procedures. We have now trimmed the text.
3- Did you determine the minimal inhibitory concentration and the minimal bactericidal concentration from combining both drugs together.
The MIC of the combined treatment was determined in Banerjee et al. Sci. Rep. 11, 8690, 2021, and the optimal combined concentration of both methicillin and AEA was found to be 50 μg/ml. Thus, this concentration was used for studying the biochemical and molecular processes affected by AEA.
4- why do not grow your bacteria for a longer time, more than 8hrs as the drugs might kill the bacteria.
The OD, CFU and ATP content were also studied after a 24 h incubation. The maximum effect of the combined treatment was observed after an 8 h incubation, with similar values after 24 h, indicating an initial bactericidal effect, followed by a bacteriostatic effect. We have added text to the Result section to clarify this point.
5- Figures 1 B, D, F. Add the controls for the figures.
Figures 1B, D, and F are the methicillin/AEA combinations shown in Fig. 1A, C and E, respectively, where the controls are included. In A, C, and E, the changes caused by the drug combination was swallowed due to the high values of control and the single treatments. In B, D and F, the Y-axis was adjusted to fit the values of the drug combination. In order to clarify this issue, we have added text to the Result section.
6- You need to keep the results only in the result section, any comparisons from previous publications can move to the discussion section with citations.
The first paragraph of each Result subsection describes the rational for why the given issue was studied, and provides the necessary background information required for understanding the data presented. Since the present study provides a mechanistic insight into the phenomena described in Banerjee et al. Sci. Rep. 11, 8690, 2021, it is unavoidable not to mention the connections.
7- In Figures 3 and 6. Is the loaded amount and concentration on the gel similar in treated and control samples?
Similar amount of control and treated samples are loaded, which is demonstrated by the silver stained gels of the same samples.
8- Give more focus on the results obtained by this study, not your previous studies.
As mentioned above, in order to understand why the different topics were studied, each Result subsection was initiated with a rational. We have rewritten some parts of the result section to better emphasize the data obtained from this study.

Reviewer 2 Report
The manuscript entitled “Targeting the Achilles’ Heel of Multidrug-Resistant Staphylococcus aureus by The Endocannabinoid Anandamide” is appropriately designed, written and structured by Sionov et al., suitable English with real clear structure. They carried out valuable research to evaluate the sensitizing effects of arachidonoylethanolamine on Multidrug-Resistant Staphylococcus aureus. The results are so interesting, they found that arachidonoylethanolamine increased the intracellular drug concentration of fluorescent penicillin and augmented its binding to membrane proteins with concomitant altered membrane distribution of these proteins. They showed the sensitization effects of arachidonoylethanolamine on Multidrug-Resistant Staphylococcus aureus. This manuscript is very valuable and strongly recommended for publication in this prestigious high-ranked journal, the International Journal of Molecular Sciences.
Author Response
We thank the Reviewer for reading and critically reviewing the manuscript.
Round 2
Reviewer 1 Report
The authors answered all the comments, however, the manuscript is still very long and has a lot of unnecessary details that can be removed especially the result and the discussion sections.
Author Response
We thank the Reviewer for reading and critically reviewing the manuscript.
We are not sure what are the unnecessary details that are intended here. The results section describes the data obtained, and the Discussion section integrates and discusses these findings in light of the scientific literature. We have again carefully read through the manuscript to look for superfluous text, and found that all text in the current version is necessary, and removing any part would reduce the quality of the paper.
We are aware that this manuscript presents a comprehensive, in-depth research on several aspects of AEA on multidrug resistant S. aureus. The research studied the effect of AEA on various biochemical pathways that have been attributed roles in antibiotic resistance and biofilm formation. Several aspects have been addressed including drug retention, PBP2a membrane distribution, EPS production, cell wall teichoic acid content, membrane fluidity, membrane cardiolipin content, membrane bound eDNA, autolysin and proteolytic activities. Additionally, the activity of membrane-bound ATPase that is important for many biochemical processes including membrane transport, was studied. It is essential to discuss all of these aspects in one manuscript and not divide them into several separate papers, since in the current format, it provides a better comprehensive understanding of the mode of action and the integrative interrelationship between these processes. As described in the manuscript, the AEA-mediated inhibition of membrane transport has not only implications on intracellular drug concentration, but also on EPS secretion, cell wall teichoic acid and consequent autolysin levels in the cell wall, altogether resulting in increased drug susceptibility, reduced biofilm formation, and growth arrest. In addition, AEA promotes spontaneous cell lysis, that also might explain their higher susceptibility to various compounds including antibiotics, methyl green and the cardiolipin-binding NAO. Since the biochemical pathways in bacteria are complex and not straight forward, the scientific descriptions are important for proper comprehension of the studied issues.

Round 3
Reviewer 1 Report
Thank you for addressing my comments.